# The genome regulatory landscape of Atlantic salmon liver through smoltification

**Thomas N. Harvey** [1]◉, **Gareth B. Gillard** [1]◉, **Line L. Røsæg** [1]◉, **Fabian Grammes** [2], **Øystein Monsen** [3], **Jon Olav Vik** [4], **Torgeir R. Hvidsten** [4], **Simen R. Sandve** [1]*

**1** Centre for Integrative Genetics (CIGENE), Department of Animal and Aquacultural Sciences, Faculty of Biosciences, Norwegian University of Life Sciences, Ås, Norway, **2** AquaGen AS, Trondheim, Norway, **3** Michael Sars Centre, University of Bergen, Bergen, Norway, **4** Faculty of Chemistry, Biotechnology and Food Sciences, Norwegian University of Life Sciences, Ås, Norway

◉ These authors contributed equally to this work.
* simen.sandve@nmbu.no

**Data Availability Statement:** Sequencing data is in the European Nucleotide database for RNA-seq (PRJEB52829), ATAC-seq (PRJEB72206), and RRBS (PRJEB60411), and also available through

## Abstract

The anadromous Atlantic salmon undergo a preparatory physiological transformation before seawater entry, referred to as smoltification. Key molecular developmental processes involved in this life stage transition, such as remodeling of gill functions, are known to be synchronized and modulated by environmental cues like photoperiod. However, little is known about the photoperiod influence and genome regulatory processes driving other canonical aspects of smoltification such as the large-scale changes in lipid metabolism and energy homeostasis in the developing smolt liver. Here we generate transcriptome, DNA methylation, and chromatin accessibility data from salmon livers across smoltification under different photoperiod regimes. We find a systematic reduction of expression levels of genes with a metabolic function, such as lipid metabolism, and increased expression of energy related genes such as oxidative phosphorylation, during smolt development in freshwater. However, in contrast to similar studies of the gill, smolt liver gene expression prior to seawater transfer was not impacted by photoperiodic history. Integrated analyses of gene expression, chromatin accessibility, and transcription factor (TF) binding signatures highlight chromatin remodeling and TF dynamics underlying smolt gene regulatory changes. Differential peak accessibility patterns largely matched differential gene expression patterns during smoltification and we infer that ZNF682, KLFs, and NFY TFs are important in driving a liver metabolic shift from synthesis to break down of organic compounds in freshwater. Overall, chromatin accessibility and TFBS occupancy were highly correlated to changes in gene expression. On the other hand, we identified numerous differential methylation patterns across the genome, but associated genes were not functionally enriched or correlated to observed gene expression changes across smolt development. Taken together, this work highlights the relative importance of chromatin remodeling during smoltification and demonstrates that metabolic remodeling occurs as a preadaptation to life at sea that is not to a large extent driven by photoperiod history.

ArrayExpress (RNA-seq: E-MTAB-11746, ATAC-seq: E-MTAB-13743). Code for running all steps of the analysis and generating results and figures is available on gitlab (gitlab.com/sandve-lab/GSFsmolt).

**Funding:** This work was supported by the projects GenSysFat (Norges Forskningsrådet 244164 to SS) and DigiSal (Norges Forskningsrådet 248792 to JOV). The funders had no role in study design, data collection and analysis, decision to publish, or preparation of the manuscript. The funder website can be found here: https://www.forskningsradet.no/.

**Competing interests:** The authors have declared that no competing interests exist.

## Introduction

Atlantic salmon are an anadromous species. They begin life in freshwater riverine habitats, then migrate to sea to grow and mature before returning to freshwater to spawn. The seawater migration is preceded by a "preparatory" process that influences a range of behavioral, morphological and physiological traits, referred to as smoltification [1]. This includes changes in pigmentation and growth [2], ion regulation [3, 4], the immune system [5], and various functions of the metabolism [6, 7].

The timing of smoltification is regulated by the physiological status of the fish [8], as well as external environmental signals such as temperature and day length [2, 9, 10]. Salmon smoltify in the spring, and the transition from short to long days is believed to drive changes in hormonal regulation and initiate smoltification. In line with this model, we recently demonstrated that exposure to a short photoperiod (i.e. a simulated winter photoperiod) induces transcription of a subset of photoperiod-history sensitive genes [3], dampens acute transcriptomic responses to increased salinity, and results in enhanced seawater growth [11]. These findings support a model of smolt development regulation, where photoperiodic-history drives genome regulatory remodeling underlying key smoltification associated phenotypes.

Although gill physiology has been most studied in the smoltification literature due to its role in osmoregulation [12], other organs such as the liver also undergo large changes in function upon smoltification and seawater migration, with large implications for key metabolic traits. The liver acts as a metabolic hub, processing metabolites obtained through the diet, transforming them, and transporting the products throughout the body. It is therefore a logical place to study metabolic shifts associated with life-stage transitions. Lipid metabolism is especially critical in fish, as natural aquatic diets are generally high in lipid content, especially long chain polyunsaturated fatty acids (LC-PUFA) [13], and Atlantic salmon are a key source of healthy long chain omega-3 fatty acids in human diets [14]. It has been shown that lipid composition in Atlantic salmon reared on diets containing high or low LC-PUFA have distinct lipid profiles during the parr stage, then during smoltification the lipid profiles converge [15, 16]. This is likely a consequence of smoltification associated increased lipolytic rates and decreased lipid biosynthesis [6, 7]. In a recent study we demonstrated large changes in lipid metabolism gene regulation across the fresh-saltwater transition following smoltification [17]. Unfortunately, in this study smolts in freshwater were not sampled, hence it remains unclear if photoperiodic history is involved in shaping the molecular phenotype of the smolt liver as we observe in gills.

In this study, we conducted a smoltification trial to test if the photoperiodic history influences the genome regulatory landscape of Atlantic salmon liver. To do this we generated transcriptome, chromatin accessibility, and DNA methylation data across smolt development and seawater transfer to characterize the transcriptomic changes in smolts reared with a short winter-like photoperiod (8:16) compared to smolts reared on constant light (24:0). We test if photoperiodic history affects the smolt liver phenotype at the level of gene expression and use chromatin accessibility data to identify putative regulatory pathways and transcription factors involved in life-stage associated changes in liver function from the juvenile stage in the freshwater environment to an adult fish in seawater.

## Results

### Gene expression changes support decreased lipid metabolism and increased protein metabolism and energy production during smoltification

The main goal was determining the effects of smoltification on metabolism and whether there was an effect of exposure to a short photoperiod (i.e. a winter) on the gene regulation in the

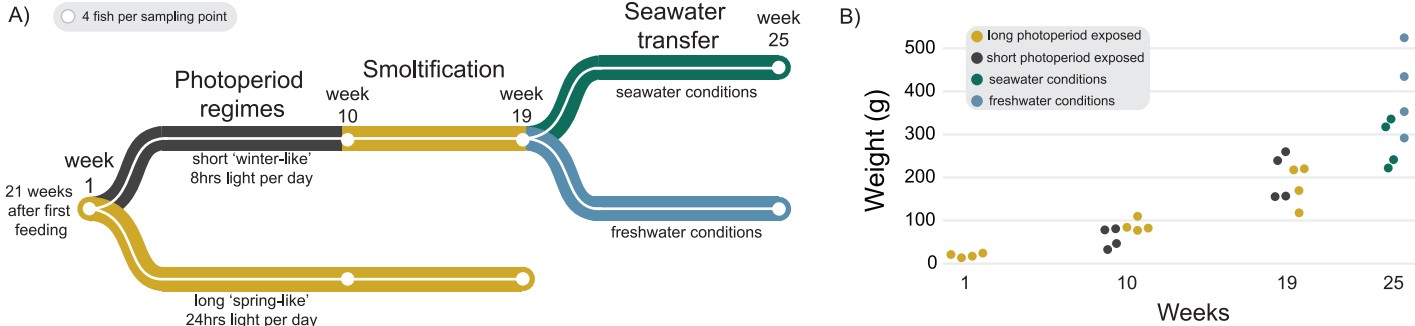

**Fig 1. Salmon growth over time.** Schematic of the experimental design and weight of salmon over time. Fish were reared for 21 weeks after first feeding in constant light conditions prior to week 1 sampling. The short photoperiod group (black, solid line) was exposed to a 8 hours light per day before being switched back to constant light and sampled at week 10. After a smoltification period, fish were sampled at week 19, then transferred to seawater conditions and sampled lastly at week 25. A long photoperiod group (grey, dashed line) received constant light throughout the experiment, and a freshwater control group branched off from the short photoperiod group by remaining in freshwater. Four fish were sampled randomly at each timepoint for RNA-seq. Two of those sampled fish were used for ATAC-seq and two additional fish for RRBS per timepoint.

liver. To accomplish this, we reared three groups of salmon for 46 weeks on commercial diets, from parr, through smoltification, and 6 weeks following transfer to seawater (Fig 1). The short photoperiod group was given an artificial winter-like photoperiod (8 hours light, 16 hours dark) for 8 weeks before they were returned to constant light, while the long photoperiod group was reared under constant light throughout the experiment. Finally, the freshwater control group contained fish from the short photoperiod group that was not transferred to sea. Following smoltification, fish transferred to seawater grew more slowly than fish that remained in freshwater (Fig 1, S1 Table). There was no mortality throughout the freshwater portion of the trial, but some mortality (8x fish) in one tank due to improper oxygenation immediately after seawater transfer.

To characterize global transcriptome changes through key life stages, under a semi-natural developmental trajectory, we sampled liver tissue from fish at each sampling point for RNA sequencing. We first tested for changes in gene expression in the cohort of fish experiencing a short photoperiod (artificial winter), followed by transfer to seawater, using an ANOVA-like test for any significant changes between any time points. This yielded 3,845 differentially expressed genes (DEGs, FDR <0.05) which were assigned to seven co-expression clusters using hierarchical clustering (Fig 2A, S2 Table). These clusters reflected patterns of gene regulatory changes (Fig 2B); peak expression levels in smolts (clusters 2 and 3), peak expression following the short photoperiod (clusters 4 and 5), decreased expression after short photoperiod and in smolts relative to all other time points (cluster 6), steady decrease in expression from parr throughout the experiment (cluster 7), and strong increase in expression in seawater (cluster 1).

To study the biological functions of genes associated with different gene expression trends, we performed KEGG enrichment analysis on each co-expression cluster, yielding 56 unique significantly enriched (adjusted p <0.05) pathways (Fig 2C). Cluster 1 genes strongly increased in relative expression after seawater transfer and was exclusively enriched in the ribosome pathway. Genes in clusters 2 and 3 that increased during smoltification and sharply decreased after seawater transfer were enriched in pathways related to genetic information processing, cell growth, and oxidative phosphorylation. Genes in clusters 4 and 5 which had peak expression after a short photoperiod and decreased during smoltification and seawater transfer contained relatively few genes and were enriched in a few carbohydrate and lipid metabolism

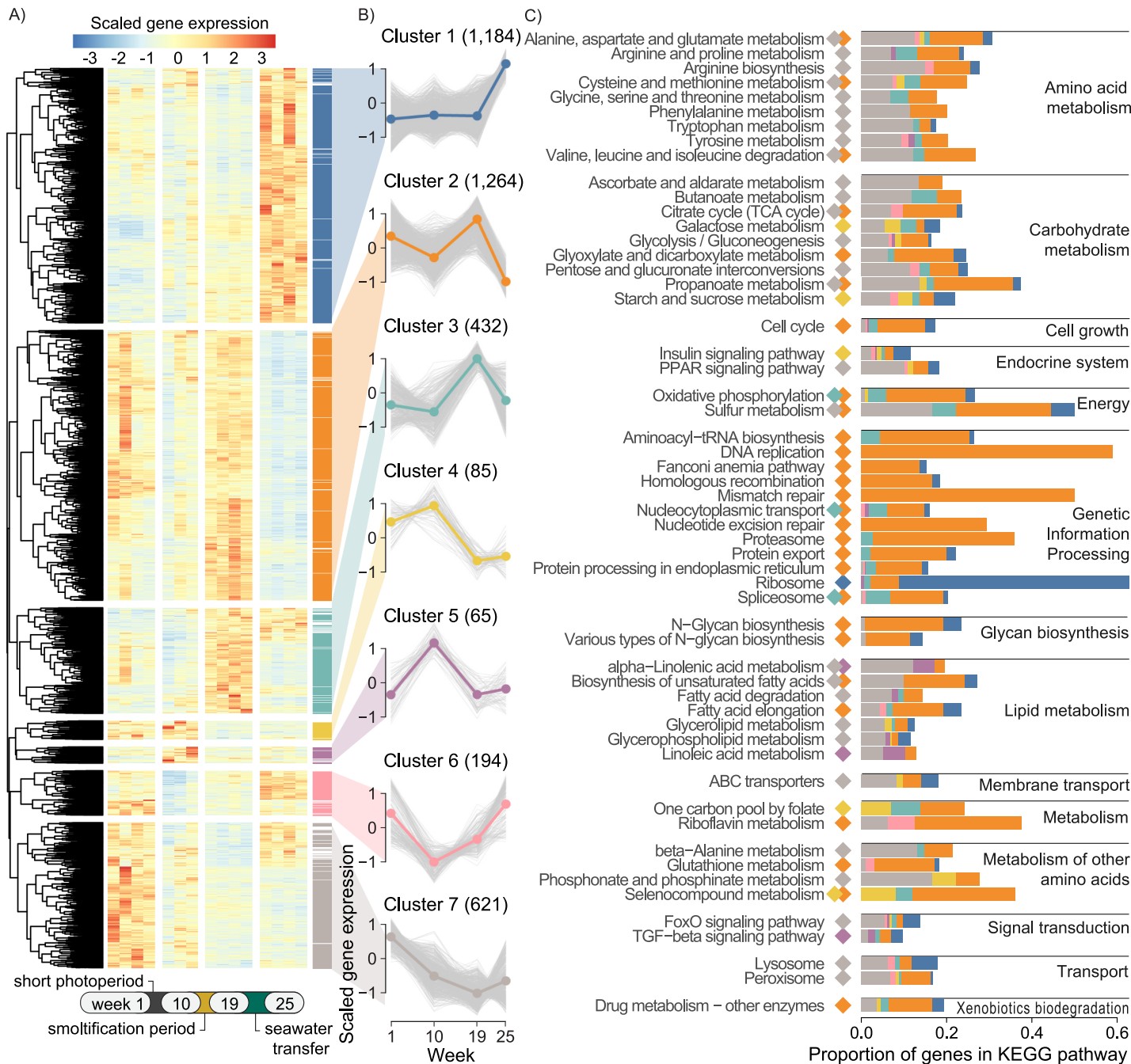

**Fig 2. Global gene expression changes across salmon life-stage.** A) Relative liver expression of genes differentially expressed between any time point in the fish cohort that experienced short photoperiod fish followed by seawater transfer (FDR <0.05). Scaled expression is denoted as gene-scaled transcripts per million. Genes were partitioned into seven co-expression clusters by hierarchical clustering. Colored bars indicate cluster membership when correlation to the mean cluster pattern was >0.5. Genes with correlation = <0.5 were excluded. B) Gene expression trends over time by cluster. Colored line indicates mean relative expression while grey lines are individual genes within the cluster. C) KEGG pathway enrichment by cluster. Colored diamonds indicate for pathways which clusters they are significantly enriched in (adjusted p <0.05). Colored bars indicate the proportion of genes within the pathways that are in clusters.

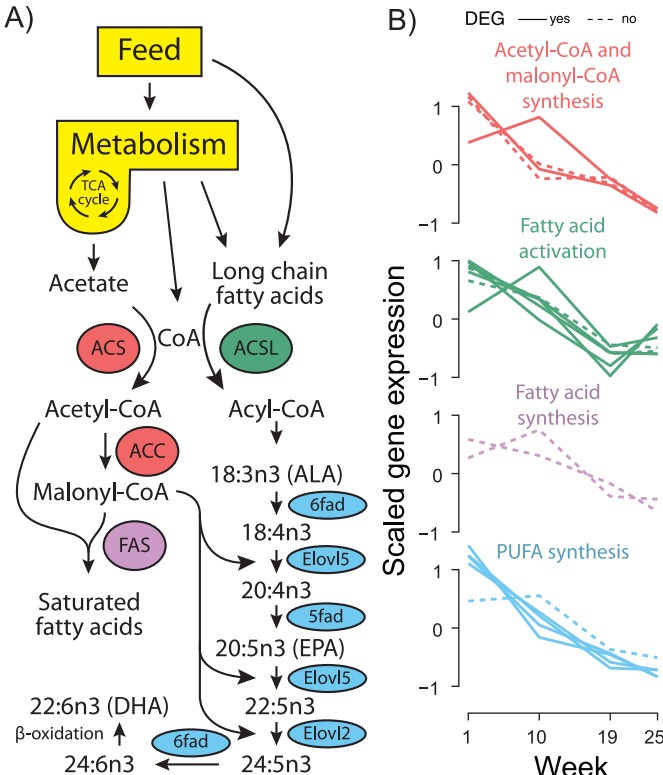

**Fig 3. Expression of key lipid metabolism genes across the parr-smolt transition.** A) Schematic of the lipid biosynthesis pathway in Atlantic salmon. B) Relative expression of genes in the pathway over time. Acetyl-CoA and malonyl-CoA synthesis (red) displays genes *acc1*, *acs2l-1*, *acs2l-2*, and *acs2l-3*. Fatty acid activation (green) displays genes *acsl1*, *acsl3l-1*, *acsl3l-2*, *acsl3l-3*, *acsl4*, *acsl4l-1*, *acsl4l-2*. Fatty acid synthesis (purple) displays gene *fas1* and *fas2*. Poly unsaturated fatty acid (PUFA) synthesis (blue) displays genes *5fad*, *6fada*, *6fadb*, *elovl2*, and *elovl5b*. Genes found differentially expressed (FDR<0.05) across life stages (Fig 2A) have a solid line and those not significant a dotted line.

pathways. Genes in cluster 6 had no significantly enriched pathways. Genes in cluster 7 which decreased in relative expression during smoltification and remained low during seawater transfer were enriched mainly in lipid, amino acid, and carbohydrate metabolic pathways, ABC transporters, and signaling pathways including FoxO signaling and PPAR signaling.

Since many KEGG pathways contain enzymes with reciprocal activities, we manually examined genes within select enriched KEGG pathways to determine what was driving enrichment trends. In lipid metabolic pathways we observed a distinct bias in genes relating to long-chain fatty acids towards downregulation in freshwater smolts. Seven long-chain-fatty-acyl-CoA ligase (*acsl*) genes (*acsl1*, three *acsl3* and three *acsl4*), acetyl-CoA carboxylase (*acc1*), three acetyl-CoA synthetase genes (*acs2l-1*, *acs2l-1*, and *acs2l-1*), several key genes related to polyunsaturated fatty acid (PUFA) biosynthesis (*5fad*, *6fada*, *6fadb*, *elovl5b*, and *elovl2*), both copies of fatty acid synthase (*fas1* and *fas2*), and three copies of the key gene diacylglycerol acetyltransferase (two *dgat1* and one *dgat2*) all significantly decreased during smoltification and remain lowly expressed through seawater transfer (Fig 3B, Table 1).

Synthesis of acetyl-CoA by *acs* and activation of long-chain fatty acids by *acsl* is the first obligatory step for entry into beta-oxidation or biosynthesis pathways (Fig 3A), so a decrease in these gene products likely means that metabolism of their substrates (acetate and C12 to C20 fatty acids) also decreases [18]. Finally, three copies of the key gene diacylglycerol

**Table 1. Key lipid metabolism gene expression fold changes.** Log2 fold change and average expression levels (cpm) of important lipid metabolism genes in salmon liver. All comparisons represent key life stages relative to freshwater parrs (week 1). These include just after shortened photoperiod (week 10 vs 1), after smoltification in freshwater (week 19 vs 1), and after seawater transfer (week 25 vs 1). Genes found differentially expressed (FDR<0.05) across life stages (Fig 2A) are marked with an asterisks (*) next to their gene name.

| NCBI gene ID | Gene name | Average CPM | Fold change (log$_2$) | | |
|---|---|---|---|---|---|
| | | | Week 10 vs 1 | Week 19 vs 1 | Week 25 vs 1 |
| 106603271 | acc1* | 315 | 0.82 | -0.93 | -3.73 |
| 106595379 | acs2l-1* | 49 | -0.4 | -0.93 | -1.33 |
| 106596660 | acs2l-2 | 182 | -0.12 | -0.62 | -0.83 |
| 106569828 | acs2l-3 | 56 | -0.56 | -0.83 | -1.33 |
| 106612015 | acsl1 | 5 | 0.12 | -1.37 | -1.47 |
| 106592274 | acsl3l-1* | 1 | -0.42 | -3.15 | -3.28 |
| 106613112 | acsl3l-2 | 1 | -0.16 | -3.32 | -2.95 |
| 106613121 | acsl3l-3* | 5 | -0.19 | -3.13 | -3.25 |
| 100380405 | acsl4* | 12 | 0.1 | -1.58 | -0.43 |
| 106603377 | acsl4l-1* | 4 | -0.38 | -1.98 | -0.78 |
| 106604299 | acsl4l-2* | 22 | 1.31 | -1.64 | -0.82 |
| 106589612 | fas1 | 399 | 0.09 | -1.11 | -4.32 |
| 106610271 | fas2 | 93 | 1.14 | -2.65 | -3.47 |
| 100136383 | 5fad* | 783 | -0.46 | -2.59 | -2.59 |
| 100136441 | 6fada* | 1453 | -0.38 | -2.66 | -3.39 |
| 106584797 | 6fadb | 7 | 0.55 | -2.09 | -3.47 |
| 100192341 | elovl2* | 171 | -0.6 | -1.31 | -1.77 |
| 100192340 | elovl5b* | 294 | -0.31 | -1.02 | -1.42 |

acetyltransferase (two *dgat1* and one *dgat2*) which catalyzes the last committed step in triacylglycerol biosynthesis [19] decreased in freshwater smolts. Collectively, co-downregulation of these important lipid associated genes is a strong indicator of decreased utilization and processing of fatty acids, especially LC-PUFA, in smolts preparing to enter a seawater environment.

We identified genes directly influenced by seawater transfer by performing a pair-wise test for gene expression changes between the short photoperiod group after seawater transfer and freshwater control group at week 25. This resulted in 2,121 DEGs (FDR <0.05, S5 Table), most (1227) being downregulated in seawater compared to freshwater (S1 Fig) and overlapped with many genes belonging to co-expression clusters 1 (281) and cluster 2 (529) in Fig 2. Regarding lipid metabolism, processes related to *de novo* fatty acid synthesis decreased in seawater relative to control. Both copies of fatty acid synthase (*fas1* and *fas2*) and one acetyl-CoA carboxylase (*acc1*) decreased expression in seawater, all of which catalyze key steps in *de novo* fatty acid synthesis [20]. Additionally, two other *acsl* genes (*acsbg2* and *acsl1*) known to be involved in saturated and monounsaturated fatty acid activation were downregulated in seawater [21]. This coincided with an increase in several thioesterase genes, including *acot1l* and *acot5l*, responsible for de-activation of fatty acids through the hydrolysis of acyl-CoAs [22]. It is unlikely that de-smoltification occurred in the freshwater control smolts because expression of these genes remains stable between weeks 19 and 25 in the control fish. This combination of decreased expression of key *de novo* biosynthesis genes and increased fatty acid de-activation through greatly increased thioesterase expression suggests salmon have a reduced capacity to synthesize fatty acids in liver after transition to sea, in line with previous findings [17].

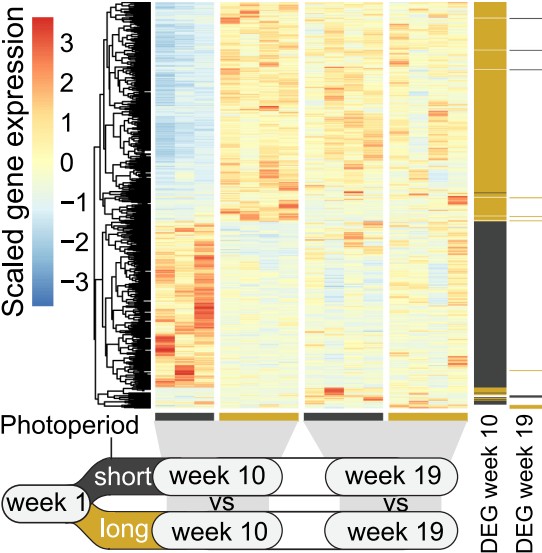

**Fig 4. Gene expression changes in response to photoperiod history.** Relative liver expression of differentially expressed genes (DEGs, FDR <0.05) between the short photoperiod group and long photoperiod group at weeks 10 and 19. Genes are marked to the right if differentially expressed higher (black) or lower (yellow) in response to a short photoperiod at weeks 10 and/or 19.

## No long-term effect of short photoperiod exposure in liver

To evaluate the role of photoperiodic history on the development of liver function during smoltification, we performed pair-wise tests for gene expression changes between fish exposed to a short photoperiod and fish on constant light regime at week 10 (at the end of the short photoperiod exposure) and at week 19 (just prior to seawater transfer). We identified a relatively shorter list of DEGs (532, FDR <0.05, S3 Table) associated with photoperiodic history differences at week 10, but only a few DEGs at week 19 (15, Fig 4, S4 Table). At week 10 we found a vitamin D 25-hydroxylase gene, the first step in the formation of biologically active vitamin D [23], was strongly downregulated after a short photoperiod (log2 fold change -7.15). This is likely due to decreased UV mediated vitamin D synthesis in the skin from less exposure to light [24]. The low number of DEGs at week 19, and low overlap between week 10 and 19 expression changes, showed that different photoperiodic histories did not impact longer term gene function in the liver following smoltification.

## Chromatin accessibility and transcription factor binding remodeled during smoltification and seawater transfer

To better understand the mechanistic drivers shaping changes in liver gene expression through salmon life stages, we generated ATAC-seq data to measure accessibility of chromatin and used this to indirectly quantify transcription factor (TF) occupancy at predicted TF binding sites (TFBS) by assessing local drops in chromatin accessibility (aka footprints). For each time point across the 25 week experiment group we generated two replicate samples of ATAC-seq data from the same livers sampled for RNA-seq, at a depth of 55-72M reads. Reads were aligned to the genome (41-63M) and peaks where reads were concentrated were defined as regions of accessible chromatin. A unified set of the ATAC peaks was made by merging peaks across the different weeks (S1 File). A principal component analysis (PCA) on the sample's read counts over the unified peaks showed pairing of replicates and separation between the

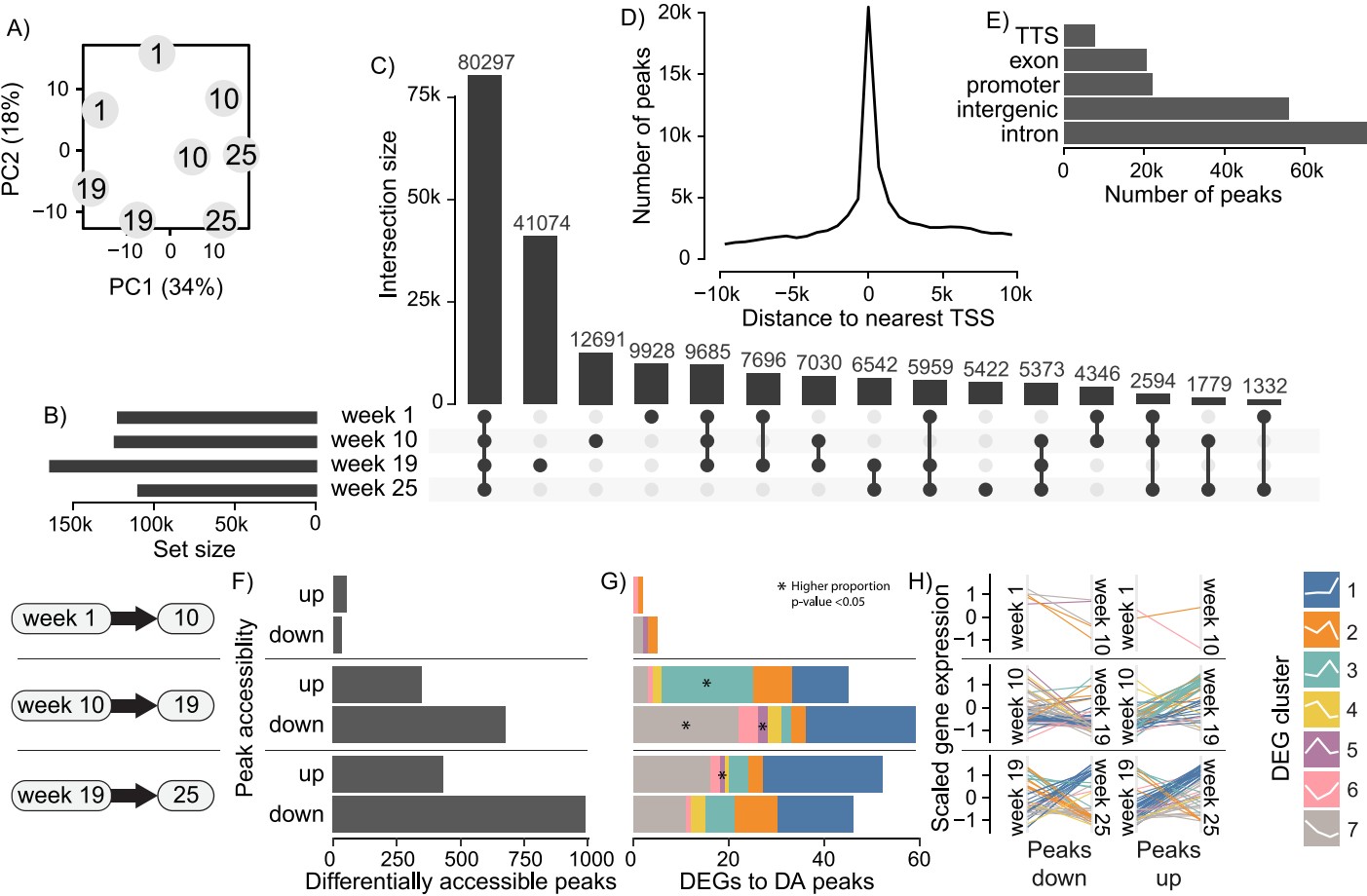

**Fig 5. Comparison of chromatin accessibility across life-stage.** A) Similarity of ATAC-seq samples at different weeks by principal component (PC) analysis of read counts within a unified set of ATAC peaks (shared and unique between weeks). B) Number of ATAC peaks called at each week. C) Number of peaks intersecting between sets or unique to each week. D) Distribution of distances (in base pairs, max absolute distance of 10kb) of unified peaks to the nearest gene transcription start site (TSS). E) Genomic locations of unified peaks. F) Number of differentially accessible (DA) peaks between key life-stages with significant (FDR <0.05) fold change up or down in reads overlapping unified peak set. G) Number of differentially expressed genes (DEGs) associated by closest proximity to the DA peaks, colored by co-expression cluster membership. Proportions of gene clusters significantly greater (p-value <0.05) between up and down DA peaks are marked by an asterisk (*), tested by Fisher's exact test. H) Expression trends across weeks of the DEGs to DA peaks down or up, colored by cluster membership.

weeks (Fig 5A). PC1 separated weeks 1 and 19 to 10 and 25, while PC2 separated the pre-smolt weeks (1 and 10) to the post-smolt (19 and 25). The unified set of 201k peaks was composed of peak sets from each week, with week 19 having the highest number of peaks (181k, Fig 5B). Most of the peaks at each week were shared across sets, with week 19 standing out as having the greatest number of unique peaks (Fig 5C). Peaks were highly concentrated around the TSS of genes as expected (Fig 5D). Peaks were mostly found in introns or intergenic regions, suggesting a higher proportion of peaks at enhancer than promoter elements (Fig 5E).

To characterize changes in chromatin accessibility across major life-stage transitions, we identified differentially accessible (DA) peaks during three key periods; shortened photoperiod (weeks 1 to 10), smoltification (weeks 10 to 19), and seawater transfer (weeks 19 to 25). Relatively few DA peaks were identified during the transition to shortened photoperiod (50 up, 32 down) relative to smoltification (349 up, 679 down) and seawater transfer (433 up, 990 down) (Fig 5F). To link these changes in chromatin structure to gene expression we associated each

DA peak to the nearest gene and counted the number of previously identified DEGs in the sets of DA peaks. Interestingly, the gene expression trends of associated DEGs largely matched the direction of DA peaks (Fig 5G). For example, during smoltification DA peaks with increased accessibility were linked to more DEGs belonging to cluster 3 (Fig 5G) which was enriched in the oxidative phosphorylation pathway (Fig 2C) and increased in gene expression (Fig 5H). Conversely, DA peaks with decreased accessibility were linked to more DEGs belonging to cluster 7 (Fig 5G) which was enriched in metabolic pathways (Fig 2C) and decreased in gene expression (Fig 5G). During seawater transfer DA peaks with both increased and decreased accessibility were highly associated with DEGs in cluster 1 which increases in expression at sea, but a much higher proportion of these were associated increased peak accessibility (Fig 5F). This implicates chromatin remodeling as an important driver of gene regulation during smoltification and seawater transfer.

Little is known about the environmentally driven changes in gene regulatory pathways in salmon. We used a TF footprinting analysis to identify within peaks drops in reads at TFBS, indicating a bound TF (i.e. occupancy) at that site in the given sample. We first describe the TFs showing genome-wide changes in TFBS binding between livers sampled in different photoperiods and water salinities (S6 Table). Since developmental stage (age) can also impact TF binding patterns, we focused on TFs with differences in occupancy that persisted across environmental contrasts with fish from different developmental stages (Fig 6A and 6B). Using a cutoff for differential TF occupancy (log$_2$ fold change in genome wide TF-motif

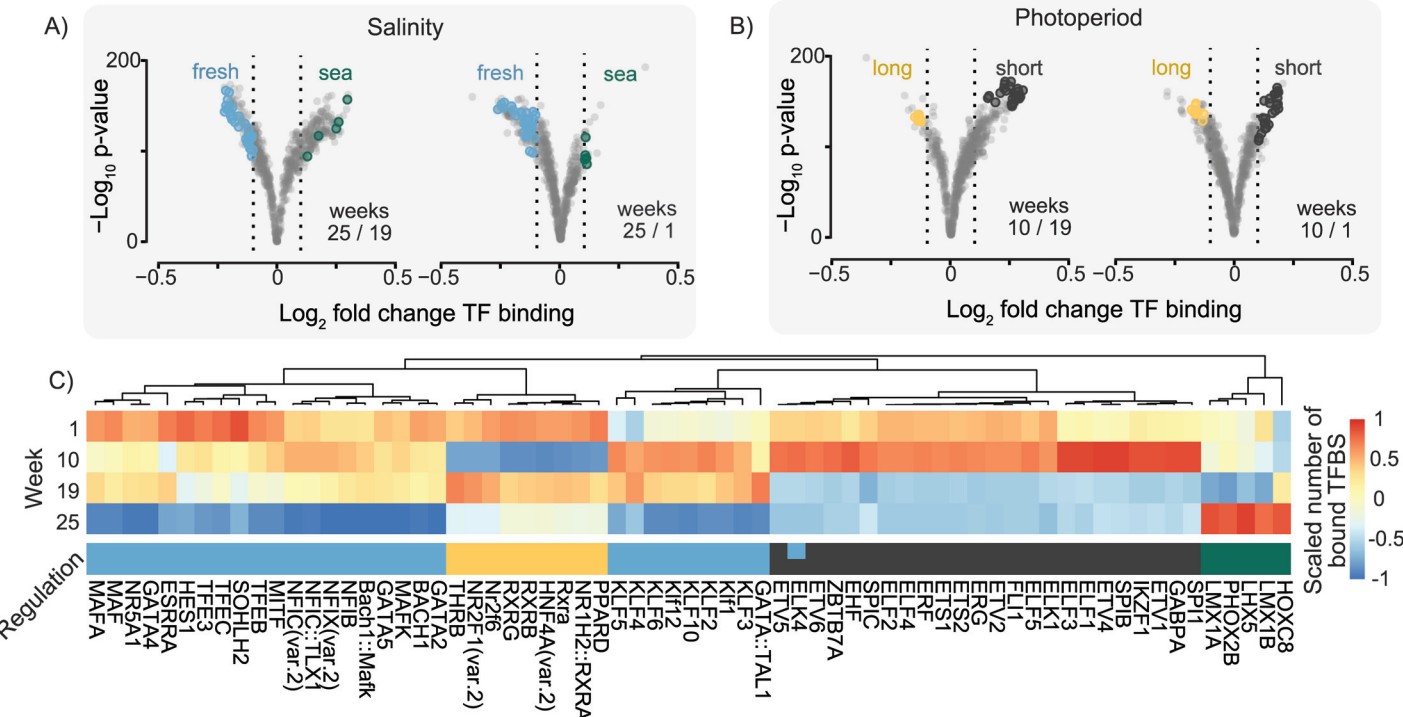

**Fig 6. Genome wide changes in transcription factor activities across key life stages.** Volcano plots show genome wide fold changes and significance for transcription factor binding site (TFBS) binding scores between A) fresh- to saltwater weeks and B) short to long photoperiod weeks. Transcription factor (TF) motifs with significant changes in global binding scores (absolute log$_2$ fold change >0.1) across each contrast are colored. C) Heatmap shows the scaled number of total bound TFBS across the weeks for the TF motifs that are significant in A) and B). The 'regulation' color indicates in which environment the TF motif had the greater TFBS binding score, freshwater (blue), seawater (green), long photoperiod (yellow), or short photoperiod (black).

occupancy >0.1) we identified 33 and 35 TF binding motifs that were associated with photoperiod and salinity, respectively (Fig 6C, S7 Table).

Most (30/35) TF motifs associated with water salinity differences were found to have a marked drop in genomic occupancy after transition to saltwater (Fig 6C). These include several motifs known to bind TFs associated in energy homeostasis related processes in mammals, such as TEF's, GATA4, NR5A1 MAF, KLFs [25–27]. Only five TF motifs had a significantly higher occupancy in saltwater, including LIM's and two homeobox binding motifs (PHOX2B and HOXC8). The photoperiod contrast (Fig 6B and 6C) revealed that most TFBS's with induced occupancy after a short day period were binding sites for E-26 family transcription factors (ETS, ERG, ETV, SPIC, ELK, SPI1), which have been associated with regulation of circadian genes in other species [28, 29]. It is interesting to note that these TF binding sites with a spike in occupancy after a short photoperiod dropped dramatically towards the end of smoltification (week 19) and stayed low after seawater transfer (week 25). TF binding sites with reduced occupancy following a period of shorter days were mostly related to homeostasis of cellular metabolism, including key liver glucose and lipid metabolism regulators PPARD [30], RXRs [31] and HNF4A [32], as well as occupancy of thyroid hormone receptor beta (THRB) [33].

Next, we tested for a link between TF binding patterns and gene regulatory dynamics. We assigned TF motifs to genes (i.e. motif-gene pairs) by closest proximity and asked if genes with a particular expression pattern (Fig 7A) were enriched for TF motifs in proximity with a corresponding pattern of TF occupancy (Fig 7B). For example, for genes in expression cluster 1 with highest expression at week 25, we expected nearby binding sites to be enriched in TF motifs that are occupied by TFs in week 25, but not the other time points. Indeed, genes in most expression clusters displayed significant enrichments of TF motifs with expected binding patterns (colored red or orange), and these signatures were quite distinct for each gene expression cluster (Fig 7C, S8 Table). The comparison for the fisher's exact tests between gene expression cluster and TF binding times to find TF motif significance is displayed in Fig 7D.

From the enrichment results (Fig 7C) we found genes in cluster 1, enriched for ribosome related functions, had nearby NFIX motifs bound more often following transition to seawater. This is a transcriptional regulator known to be involved in ribosome biogenesis [34]. In addition, cluster 1 had FOS and JUN motifs bound more after seawater entry. These TFs are major components of the Activator Protein 1 (AP-1) transcription factor complex which is responsive to growth factors and drive cell proliferation and differentiation [12, 35, 36]. The top associated TFs for cluster 2 genes were ZNF341 known to be involved in immune homeostasis [37] and several Fox TFs (A, F, L, I, K) linked to various cell physiological processes [38, 39]. Cluster 3 genes were associated with several unnamed zinc finger transcription factors (ZNFs and ZBTBs) binding at week 19 prior to seawater transfer, as well as binding of RREB1 and EGR1 TFBS in week 19 and week 25. Among TFs with binding in week 19 only, we found genes linked to regulation of immune cell function (ZBTB32 and ZNF263) [40, 41] as well as oxidative phosphorylation [42]. RREB1 and EGR1 are well described players in RAS signaling pathways [43–45] involved in cell growth and proliferation. Among cluster 4 genes we found enrichment of insulin and sugar metabolism functions, with PKNOX1 and NFYB TF binding significantly associated with their expression patterns. Both these TFs have been shown to function in lipid metabolism and be linked to insulin signaling [46–48]. The top TF associated with cluster 6 gene expression was NR1D1 (also called Reverbα), a core component of the circadian clock and regulator of lipid metabolism [49]. Cluster 6 was not enriched for any KEGG pathways but had a marked drop in expression after the short photoperiod exposure. In the final cluster 7, enriched for genes playing roles in amino acid, glucose, and lipid metabolism, we found very strong associations with binding of several TFs, including KLF and SP family

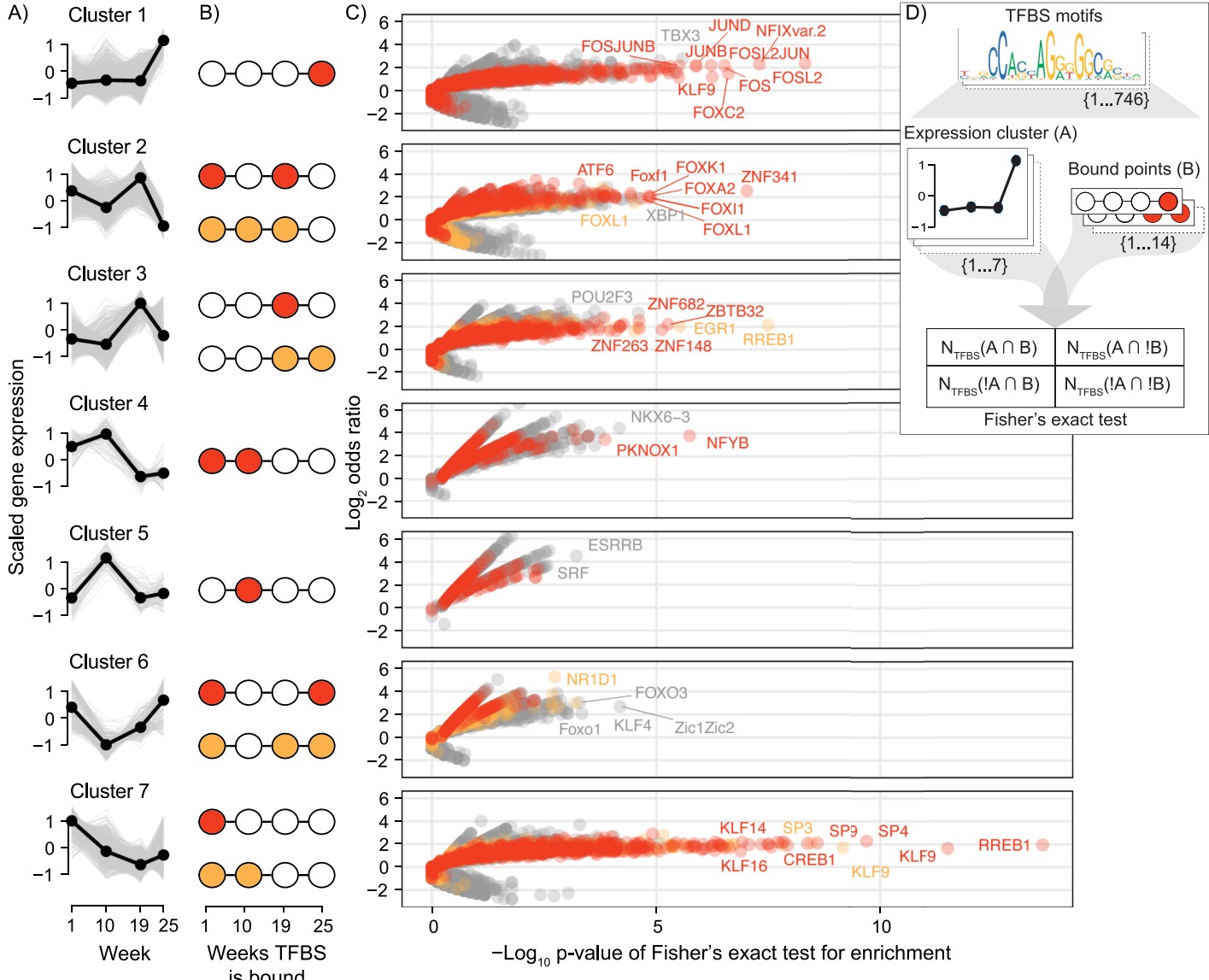

**Fig 7. Enrichment of transcription factor binding patterns within gene expression clusters.** A) Gene expression trends of clusters from Fig 1. B) Assumed weeks the transcription factor binding sites (TFBS) near genes in the cluster would be bound by a transcription factor (TF) to regulate transcription at the weeks of highest expression. A primary and secondary assumed binding pattern is colored red and orange respectively. C) For each set of genes in a cluster, each TF was tested if the TFBS near to those genes were enriched for any combination of binding pattern. Fisher's exact test results for all TF-binding pattern combinations are plotted per cluster as odds ratios against the significance. The test results for the assumed primary and secondary binding patterns in B) are colored red and orange, respectively. A top proportion of the most significant TFs (in a top quantile per test) are labeled. D) Diagram showing how each combination of TFBS motif, expression cluster, and binding pattern was tested for enrichment with a Fisher's exact test.

members. Indeed, these TFs are known to play important roles in regulating gluconeogenesis and lipid and amino acid metabolism in mammalian livers [27, 50].

## DNA methylation not linked with gene expression

To investigate if DNA methylation played a role in the gene expression changes throughout salmon life-stages, we produced a RRBS dataset from the liver samples of fish sampled concurrently with the fish used in the previous analysis at a depth of 26–40 M reads. A consensus set

of 1.2M CpG positions was used for differential methylation analysis, these were the CpGs in chromosomes with read coverage greater than 10x and present in 3 of 4 replicates. To assess genome wide differences in the regulation of CpG methylation, a principal component analysis (PCA) separated samples based on the methylation levels across the CpG consensus set. There was no clear separation of samples by timepoint or experiential condition, with PC1 and PC2 each explaining less than 5% of the variance (S3A Fig). To find specific sites of differential methylation, we tested all CpGs for differences in methylation score across any timepoints of the short photoperiod group with an ANOVA-like approach (S9 Table). Out of the consensus set of CpGs, 2535 (0.2%) were differentially methylated cytosines (DMC) across life-stages (FDR <0.05, fold change >25%, S3D–S3F Fig). Of these, 209 were present in promoter regions, 103 in exons, 664 in introns, and 782 in intergenic regions (S3E Fig). Most genomic regions with a DMC had one differentially methylated site and only a few regions contained longer stretches (up to 26 bp in an intron) of differentially methylated CpGs (Fig 3D). We assessed if these CpGs were associated to genes with a specific function, but enrichment tests for GO or KEGG terms gave no significant results. Next, we looked at if the methylation percentages of DMCs correlated to changes in their corresponding gene's expression across time-points. We identified 157 DMC-gene pairs that were significantly correlated (p <0.05, Pearson correlation coefficient >0.95), however most of these genes were relatively lowly expressed. Simulating random DMC-gene pair correlation values found that the distribution of values from our real data was not significantly different from that of simulated pairs (p-value <0.62, S3G Fig), refuting strong links between regulation of CpG methylation and gene expression in our experiment.

## Discussion

### Metabolic remodeling as a pre-adaptation to life at sea

An important feature of smoltification is the preparation of juvenile fish for a life at sea, i.e. physiological necessities for survival are already present while the fish is still in freshwater. This is well documented for the osmoregulatory machinery in the gills [51–56]. For example, the likely causal agent for saltwater tolerance, Na+/K+ ATPase α1b isoform (NKA a1b) ion transport pump, increases in abundance in freshwater gill tissues [8]. While lipid metabolism related gene expression is known to decrease in liver of seawater stage Atlantic salmon [17], this is the first report that systemic downregulation of amino acid and carbohydrate metabolism occurs in addition to lipid metabolism, and this trend actually occurs before transition to sea (Figs 2 and 3). Given the marked shift in dietary trends from mostly insect based in freshwater to marine prey in seawater [57] and the availability and nutrients such as polyunsaturated fatty acids in seawater environments is higher than freshwater [58], it is likely that the observed decrease in metabolic gene expression is a genetically programmed preadaptation to life at sea. This study therefore expands smoltification associated metabolic remodeling to include amino acid and carbohydrate metabolic pathways; and adds another feature to the list of pre-adaptations in freshwater smolts.

### The effect of photoperiod history on genome regulation during smolt liver development

Decades of research have revealed that many features of smolt physiology development can be affected by photoperiodic history [3, 59–66], including gene expression patterns. For example, a recent study of gill transcriptomes identified a subset of 96 genes with significantly increased gene expression levels in smolt exposed to a short photoperiod (8:16) during development

compared to smolts kept on constant light (24:0) [3]. In this study, however, no long-term effects of short photoperiod exposure were found in liver transcriptomes (Fig 4). We do, however, show that salmon liver gene regulation is responsive to variation in photoperiods. Transcriptome profiles through smolt development show distinct gene sets with increased and decreased expression after reduction in photoperiod (Fig 2A clusters 5 and 6). Furthermore, analyses of TF binding dynamics (Fig 6) identify photoperiod sensitive TFs encoded by genes known to have repressed expression under short photoperiod in mammals, such as retinoic acid related TFs (RXRs) and thyroid hormone receptors (THRB) [67, 68]. In addition, our integrated analyses of gene expression profiles and TF binding (Fig 7) associated NR1D1, a core component of the mammalian circadian clock [49], with genes having lower expression after exposure to short photoperiod. Taken together, we conclude that smolt liver development does not seem to rely on having experienced a winter-like photoperiod. Yet since acute effects of reduced photoperiods had a large impact on gene regulatory networks related to metabolism (Figs 6 and 7), it is likely that highly divergent photoperiodic histories can lead to delayed spillover effects and result in differences in metabolic states.

## Linking genome regulatory layers to understand the developing smolt liver

Salmon experience changes in gene regulation [16, 17] and function [6, 7] in liver during smolt development in freshwater, and following seawater entry. Yet, to our knowledge no genome wide studies of DNA methylation, chromatin accessibility, and TFs involved in driving these transcriptional and physiological changes have been published. Here, we generated an RRBS dataset as well as an ATAC-seq dataset across smolt development in liver and used the latter to predict TF occupancy dynamics and map out putative major regulators of key developmental processes during smoltification (Figs 6 and 7).

Firstly, we showed that dynamic DNA methylation has a limited role in gene regulation in the liver during smoltification. Of the 1.17 M CpGs in our dataset, only about 2500 of these showed dynamic methylation during smoltification, and few of these were in the vicinity of differentially expressed genes. This echoes an earlier study on methylation changes associated with early maturation in Atlantic salmon in which the liver exhibits less dynamic methylation overall than brain and gonads [69]. Also in other organs of Atlantic salmon, gene expression changes seem to be controlled by other gene regulatory features than DNA methylation [70]. Despite the intriguing hypothesis of DNA methylation being an important gene expression regulator, and an epigenetic one at that, our study questions this role in the context of post-embryonic development. Indeed, many studies describe methylome changes during metamorphosis or other post-embryonic transitions but do not provide strong evidence for a causative connection between changes in gene expression and changes in DNA methylation [71–73].

Previous studies of liver physiology during parr-smolt transformation highlight decreased lipid and glycogen biosynthesis and increased levels of glyco- and lipolysis [7]. In line with this, about 600 genes enriched for lipid, carbohydrate, and amino acid metabolism related functions displayed a clear decreasing trend in gene expression from parr (week 1) to smolt (week 19) (Fig 2, cluster 7). This coincided with a decrease in peak accessibility of regions near many of these genes. This points to chromatin remodeling as a driving force in pre-adaptive metabolic remodeling of salmon liver. Chromatin remodeling in salmon liver has also been implicated in sexual maturation, with strong links to changes in gene expression [70]. Several TFs showed highly significant TFBS binding associations with the cluster 7 gene expression profile (Fig 7C) including KLF/SP gene family members known to play important roles in regulating gluconeogenesis and lipid and amino acid metabolism in mammalian livers [27, 50]. Finally, genes in expression cluster 4, also showing a marked decrease in expression in smolts

(week 19), were enriched for TFBS that had a significant drop in NFY binding from week 19. This TF is known to be a major regulator of lipid metabolism, including biosynthesis of fatty acids [46]. Concurrently, but with opposite expression trends, genes involved in oxidative phosphorylation related functions (the last step in breaking down amino acids, lipids, and carbohydrates to energy) increased in expression from parr to smolts (Fig 2, cluster 3). The TFBS of these same genes were enriched for binding of the TF ZNF682 in smolts in week 19 (Fig 7C), a nuclear encoded TF gene that regulates oxidative phosphorylation in human cells [42]. Together, these results suggest that increased ZNF682 occupancy in combination with reduced KLF and NFY promoter binding has an associated link to the liver metabolic shift from synthesis to break down of organic compounds as fish undergo parr-smolt transformation.

Following the parr-smolt transformation, the transition to a life in seawater is also known to be associated with additional changes in physiology in Atlantic salmon. Genes increasing in expression in seawater (Fig 2, cluster 1) were involved in ribosome biogenesis and their TFBS were associated with increased NFIX occupancy in seawater (Fig 7C), reported to impact ribosome biogenesis in mammals [34]. Another well known route to increased ribosome gene expression and protein synthesis is the induction of the mTOR pathway [74, 75]. Interestingly, seawater entry is known to trigger increased growth hormone levels in salmon [76, 77] and this hormone acts as a rapid activator of protein synthesis through the mTOR pathway [74]. Furthermore, seawater growth hormone increase can also be linked to the second group of TFs putatively involved in gene expression induction after seawater transition (Fig 2), namely JUN and FOS (Fig 7C). These genes, originally known as onco-genes, are also responsive to growth hormones, and regulate cell proliferation and differentiation [12, 35, 36]. These TFs provide the molecular basis for linking growth hormones to increased growth capacity of smolt in seawater [4]. In our experiment freshwater control fish were larger than fish transferred to sea, but this can be explained by a known initial suppression in growth and feeding followed by increased growth rates [78]. Finally, genome-wide footprint signals (Fig 6) also pointed to large changes in the binding of TFs involved in energy homeostasis after seawater entry [25–27], further underpinning the metabolic gear shift. In conclusion, our data suggested that the genome regulatory dynamics in smolt livers across the fresh- to seawater transition is likely driven to a large extent by an increase in circulating growth hormone, resulting in activation of major regulatory pathways (e.g. JUN/FOS) for cell growth and differentiation.

Although our observations on regulatory dynamics during smoltification in Atlantic salmon present a coherent story, it is important to note that these data are correlative in nature. Future work is needed to follow up on the hypotheses generated here using formal experimental approaches.

## Conclusion

We confirm previous findings that lipid metabolism related gene expression is reduced in liver of Atlantic salmon after seawater transfer and provide the first evidence that this remodeling occurs as a pre-adaptation to life at sea, after smoltification while salmon are still in freshwater. This trend was associated with a decrease in chromatin accessibility near many associated genes and a shift in TFBS occupancy of known metabolic regulators in the KLF/SP gene family, underscoring the importance of chromatin and TF dynamics in preparatory metabolic remodeling. We find that shortened photoperiod did induce acute changes in gene expression, however these changes did not persist after the fish had been switched back to long photoperiod. This indicates that photoperiod history is unlikely to play a major role in smoltification of salmon liver, unlike studies in gill. We observed a number of TFBSs with differential occupancy during shortened photoperiod including increased occupancy of known regulators of

circadian rhythm (ETS, ERG, ETV, SPIC, ELK, SPI1) and decreased occupancy of metabolic regulators (PPARD, RXRs, HNF4A). After seawater transfer, we observed a large increase in expression of genes related to ribosome biogenesis that was associated with increase occupancy of the known mammalian ribosomal regulator NFIX. Additionally, TFBSs for growth hormone responsive TFs JUN and FOS displayed increased occupancy in seawater, providing a link between increased circulating growth hormone and increased growth capacity of smolts at sea. In general, we found that chromatin accessibility and TF occupancy patterns were closely linked to changes in gene expression. On the other hand, differential DNA methylation patterns did not have any clear association to gene expression, indicating that smoltification of liver is more dependent on chromatin and TF regulatory network remodeling than epigenetic remodeling by DNA methylation. This work is a first step in understanding the underlying molecular mechanisms of metabolic remodeling in preparation for life at sea and provides a framework for future studies to further unravel the complex regulome of Atlantic salmon.

## Materials and methods

### Smoltification trial

Atlantic salmon eggs, provided by AquaGen Breeding Centre Kyrksæterøra, Norway, were sterilized at the Norwegian University of Life Sciences (NMBU) fish lab and incubated at 350 to 372 day-degrees until hatching. First feeding of fry was five weeks after hatching when the egg sac had been depleted. Fry were reared in two replicate tanks and on a standard commercial diet high in EPA and DHA fats for the duration of the trial. Fish occasionally needed to be euthanized as they grew to maintain adequate dissolved oxygen levels in the tanks. Sampling began 21 weeks after first feeding, here called week 1, and again at weeks 10, 19, and 25. Sampled fish were euthanized by a blow to the head and samples of liver tissue were cut into ~5 mm cubes, placed in RNAlater, and incubated for at least 30 minutes at room temperature before long-term storage at -20˚C. One week after the first sampling, some fish from each tank were transferred to long photoperiod tanks where the day length remained at 24 hours of light a day throughout the experiment. At the same time, the original tanks' photoperiod was switched to short photoperiod, "winter-like", lighting conditions with 8 hours of light per day for 8 weeks to trigger smoltification before returning to "spring-like" conditions with 24 hours of light per day. Immediately after the week 19 sampling, some fish from each short photoperiod group tank were transferred to seawater conditions at the Norwegian Institute for Water Research (NIVA), Solbergstranda, Norway. UV-sterilized seawater used in this life-stage had a salinity of 3%-3.5% and was obtained from the Oslofjord. Fish were sedated before transport and allowed to acclimatize for several hours before being slowly introduced to the new water conditions. The fish that remained in freshwater were sampled at the same time as the seawater fish at week 25. All animals used in this study were handled in accordance with the Norwegian Animal Welfare Act of 19th June 2009.

### RNA sequencing

For RNA sequencing we extracted total RNA of liver samples from short phoperiod, long photoperiod, and freshwater control groups taken on weeks 1, 10, 19, and 25 in replicates of four with the RNeasy Plus Universal Kit (QIAGEN). Concentration was determined with a nanodrop 8000 spectrophotometer (Thermo Scientific) and quality was assessed by running on a 2100 bioanalyzer using the RNA 6000 Nano Kit (Agilent). Extracted RNA with an RNA integrity number (RIN) of at least eight was used to make RNA-seq libraries using the TruSeq Stranded mRNA HT Sample Prep Kit (Illumina). Mean length and library concentration was determined by running libraries on a 2100 bioanalyzer using a DNA 1000 Kit (Agilent). RNA-

seq libraries were sequenced by the Norwegian Sequencing Center (Oslo, Norway) on an Illumina HiSeq 4000 using 100 bp single end reads and at a depth of 25-43M reads per sample.

## Gene expression quantification

Gene expression was quantified from RNA-seq fastq files through the nf-core rnaseq pipeline (v3.9, 10.5281/zenodo.7130678), which involves quality control, read trimming and filtering, alignment of reads to the Atlantic salmon genome and gene annotations (NCBI refseq 100: GCF_000233375.1_ICSASG_v2) with STAR aligner, and read quantification from alignment with the salmon program. See pipeline documentation for further details on all steps: nf-co. re/rnaseq. Gene level counts, length scaled, were used for differential expression testing, and gene transcript per million (TPM) values used for visualizations including expression heatmaps and line plots. A PCA of gene TPMs showed one sample at week 10 (week_10_2_3, Biosample: SAMEA14383461) as an outlier (S2 Fig) so we removed this sample from the analysis.

## Differential expression analysis

Differentially expressed genes (DEGs) were tested for differences across all short photoperiod group time points using an ANOVA-like test with the edgeR R package (v3.36) [79], using the generalized linear model and quasi-likelihood F-test function (glmQLFTest), testing for differences between weeks 1, 10, 19, and 25. DEGs were found using an FDR cutoff of $<0.05$. Euclidean distances of DEG were calculated based on TPM values over the samples (R function dist), and the DEGs separated into 7 clusters. We chose 7 clusters based on observing patterns in the heatmap of expression and comparing the sum of squares within and between different numbers of clusters. DEGs had to have correlation $>0.5$ to the mean expression values of their assigned cluster, otherwise they were excluded from the cluster. Differential expression analysis was also performed between short and long photoperiod groups at week 10 and 19 separately, in pair-wise exact tests with edgeR (v3.36), choosing DEGs with an FDR cutoff of $<0.05$. Enrichment of KEGG pathways in sets of DEGs was performed with the clusterProfiler R package (v4.2.2), using the pathway data for Atlantic salmon genes within the KEGG database.

## ATAC sequencing

The protocol for the ATAC assay was based on that in Buenrostro et al. 2013 [80]. Two replicate liver tissue samples were used from the short photoperiod group on weeks 1, 10, 19 and 25. The liver tissues were sampled then immediately washed and perfused with cold PBS to remove blood before being dissociated and strained through a cell strainer. The nuclei were isolated from cell homogenate by centrifugation and counted on an automated cell counter (TC20 BioRad, range 4–6 um). Transposition of 100k (weeks 1, 19 and 25) and 75k (week 10) nuclei was performed by Tn5 transposase from Nextera DNA Library Preparation kit. The resulting DNA fragments were purified and stored at -20˚C. PCR Amplification with addition of sequencing indexes (Nextera DNA CD) were done according to Buenrostro et al. 2015 [81], with a test PCR performed to determine the correct number of amplification cycles. The ATAC libraries were cleaned by Ampure XP beads and assessed by BioAnalyser (Agilent) using High sensitivity chips. Quantity of libraries were determined by using Qubit Fluorometer (Thermo). Mean insert size for the libraries was 190 bp. Sequencing was done on a HiSeqX lane using 150 bp paired-end reads and at a depth of 55-72M reads per sample.

## ATAC peak calling and differential accessibility analysis

Calling of ATAC-seq peaks was done through the nf-core atacseq pipeline (v1.2.1,10.5281/zenodo.3965985), which involves quality control, read trimming and filtering, read alignments to the Atlantic salmon genome (BWA aligner), and calling of narrow peaks per sample as well as a unified narrow peak set across all samples (MACS2 peak-caller). Data for QC results including PCA of samples, intersection of peaks sets, peak distances to TSS, and peak genomic locations, were also obtained from the pipeline. See pipeline documentation for further details on all steps: nf-co.re/atacseq. Additionally read counting and differential accessibility analysis results were obtained from the pipeline. This involved DESeq2 to compute log2 fold change values between the different weeks for the unified peak set, and the files for significantly differential peaks with FDR cutoff of <0.05 were used. We assigned differential accessible peaks to genes by minimal distance of peak to gene TSS, and filtered peaks assigned to the set of DEGs determined using the ANOVA-like tests across all weeks, and their corresponding expression cluster membership.

## TF footprinting

TF footprinting of the unified ATAC peak set was completed using the TOBIAS program (v7.14.0) [82]. With it, we identified 'footprints' (i.e. dips in read depth within peaks indicative of TF proteins binding to the DNA and locally blocking transposase activity) using the ATAC-seq read alignment BAM files for each week (reads from replicates combined). We used a set of TFBS motifs from the JASPAR database (2020 CORE vertebrates non-redundant) to identify TFs associated with these footprints. Peaks were associated to a gene by closest proximity to the TSS during the ATAC-seq pipeline. Simple repetitive regions in the genome were identified and masked from the analysis using RepeatMasker (repeatmasker.org). Data for the genome-wide changes in TF binding was taken from the 'bindetect_results.txt' file produced by TOBIAS, plotting the change in TF binding scores between weeks against their p-values. The number of bound sites for each TF was scaled across weeks and used for the heatmap visualization of differences. We identified TFs changing in response to salinity or photoperiod by intersecting TF binding results between the different weeks. With a $\log_2$ fold change cutoff of >0.1, TFs that had an increase in binding in week 25 compared to both weeks 1 and 19 were assigned as 'sea', and inversely those with a decrease assigned as 'fresh'. Similarly for photoperiod those increasing in week 10 compared to 1 and 19 were assigned 'short' and those decreasing were assigned 'long'. We tested if certain TF binding patterns (which weeks TFBS were bound) were enriched within sets of genes assigned to the expression clusters previously identified. The peak-gene annotation data mentioned previously was used to assign the TFBS to genes. We used Fisher's exact tests to test for a significant enrichment of a specific TF binding pattern a TFBS motif has within a set of genes in an expression cluster. This test was done on each TFBS motif (746 motifs), for each possible binding pattern (14 patterns) and within each expression cluster (7 clusters) (see Fig 7D for a diagram of the test). Results for binding patterns that are unchanged across weeks, i.e. bound at all weeks or no weeks, were not shown in the results.

## Reduced representation bisulfite sequencing

Livers from four fish per time point (three fish for week 25) were sampled and liver tissue was stored on RNAlater at -20˚C. These were not the same fish used for RNAseq and ATACseq, but they were sampled at the same time those fish. The samples were processed with Ovation RRBS Methyl-Seq System (NuGen) and bisulfite treatment was done with the Epitect Fast Bisulfite Conversion kit (Qiagen). RRBS libraries were quality controlled with a BioAnalyser (Agilent) machine on DNA1000 chips. Paired-end sequencing was performed by Novogene

with a HiSeq X sequencing (Illumina). The mean library insert size was 168 bp and read depth was 26-40M reads.

## Alignment of bisulfite-treated reads and cytosine methylation calls

Quality trimming of reads was done with Trim Galore (v0.6.4) [83] and adapters were removed with cutadapt (v2.7) [84]. Bismark Bisulfite Mapper (v0.22.3) [83] was run with Bowtie 2 [85] against the bisulfite genome of Atlantic salmon (ICSASG_v2) [86] with the specified parameters: -q—score-min L,0,-0.2—ignore-quals—no-mixed—no-discordant—dovetail—maxins 500. Alignment to complementary strands were ignored by default. About 40–50% of the reads were mapped to the genome. Methylated cytosines in a CpG context were extracted from the report.txt-files produced by the Bismark methylation extractor. The resulting coverage files containing methylated and unmethylated CpG loci for each sample was first filtered for known SNPs in the salmon genome then used in the analysis of differential methylation. Coverage distribution around transcription start sites (TSS) and 20 kb upstream and downstream showed that the highest coverage of reads was found nearby TSS (S3B Fig) indicating that MSPI digestion of CCGGs have resulted in enrichment around TSS, as expected by the RRBS method. Using genome annotation information, we classified CpGs according to their genomic context (S3C Fig).

## Differential methylation analysis

Samples were first organized with the R package methylKit (v1.9.4) [87] and the CpG loci were filtered by read coverage, discarding those below 10 reads per locus or more than 99.9th percentile of coverage in each sample, and those not to chromosomes. CpG loci between replicates were merged, keeping those present in at least three of the samples. The differential methylation analysis was done with an ANOVA-like analysis test of edgeR [79], contrasting the counts of methylated reads at different time points. Differentially methylated CpGs (DMC) were called with an FDR <0.05. A heatmap of the differentially methylated CpGs shows row scaled methylation percentage values of DMCs (S3F Fig).

## Supporting information

**S1 Fig. Gene expression changes in response to salinity.** Relative liver expression of genes differentially expressed (FDR <0.05) between the seawater exposed short photoperiod group and freshwater control group at week 25. Genes are marked to the right if differentially expressed at week 25, green if higher in seawater, blue if higher in freshwater.
(EPS)

**S2 Fig. Principal component analysis of gene expression differences between samples.** A) Similarity between all samples in the study, by principal components (PC) of gene expression. Short photoperiod group samples are colored red, and long photoperiod group samples for the same time points are colored gray. Labeled are potential outlier samples. B) Effect of removing sample 'week_10_2_3' from the PCA.
(EPS)

**S3 Fig. Methylated CpGs from RRBS assay.** A) Similarity of samples by principal components (PC) of the read coverage of consensus CpGs. B) Genomic context of the differentially methylated CpGs. C) Heatmap of methylation values of differentially methylated CpGs during the smoltification trial. D) Density of correlation values between differentially methylated CpGs and gene expression levels, and density of correlation levels between random

gene-CpG pairs.
(EPS)

**S1 Table. Phenotypic data of sampled Atlantic salmon.** Phenotype data of Atlantic salmon used in this study. Columns provide: Unique fish identifier (*Fish ID*), week fish was sampled (*Week #*), tank number (*Tank #*), fish number (*Fish #*), date fish was sampled (*Date sampled*), fish weight (*Weight (g)*), fish length (*Length (mm)*), fish sex (*sex (M/F)*).
(TSV)

**S2 Table. Differentially expressed genes across smoltification.** Atlantic salmon genes differentially expressed (FDR <0.05) between any time points across the smoltification experiment. Columns provide: the NCBI id for genes (*gene_id*), available gene name (*gene_name*), description of coded protein (*description*), and expression cluster number genes were assigned to (*deg_cluster*).
(TSV)

**S3 Table. Differentially expressed genes in response to photoperiod at week 10.** Atlantic salmon genes differentially expressed (FDR <0.05) between week 10 short photoperiod samples, and week 10 long photoperiod samples. Columns provide: the NCBI id for genes (*gene_id*), available gene name (*gene_name*), description of coded protein (*description*), $log_2$ fold change in expression of short versus long photoperiod values (*logFC*), average expression across samples in $log_2$ counts per million (logCPM), p-value of the differential expression test (*PValue*), false discovery rate adjusted p-value (*FDR*), and the time point that was tested, in this case week 10 (*week*).
(TSV)

**S4 Table. Differentially expressed genes in response to photoperiod at week 19.** Atlantic salmon genes differentially expressed (FDR <0.05) between week 19 short photoperiod samples, and week 19 long photoperiod samples. Columns provide: the NCBI id for genes (*gene_id*), available gene name (*gene_name*), description of coded protein (*description*), $log_2$ fold change in expression of short versus long photoperiod values (*logFC*), average expression across samples in $log_2$ counts per million (logCPM), p-value of the differential expression test (*PValue*), false discovery rate adjusted p-value (*FDR*), and the time point that was tested, in this case week 19 (*week*).
(TSV)

**S5 Table. Differentially expressed genes in response to seawater transition.** Atlantic salmon genes differentially expressed (FDR <0.05) between week 25 short photoperiod samples after transition to seawater conditions, and week 25 freshwater control samples. Columns provide: the NCBI ID for genes (*gene_id*), available gene name (*gene_name*), description of coded protein (*description*), $log_2$ fold change in expression of seawater versus freshwater values (*logFC*), average expression across samples in $log_2$ counts per million (*logCPM*), p-value of the differential expression test (*PValue*), false discovery rate adjusted p-value (*FDR*).
(TSV)

**S6 Table. Global changes in transcription factor binding.** Results from TOBIAS ATAC-seq footprinting of transcription factor binding sites (TFBS) in the Atlantic salmon genome, showing the global changes in transcription factor (TF) binding for all TF motifs tested, across the different time points of the experiment. Columns provide: output file prefix of TF name with motif ID (*output_prefix*), TF name (*name*), motif ID (*motif_id*), name of the TF's cluster group (*cluster*), total number of TFBS (*total_tfbs*), columns for the mean score of TF binding across all TFBS for each time point (columns *week_1_mean_score* to *week_25_mean_score*),

total number of bound TFBS for each time point (columns *week_10_bound* to *week_25_-bound*), and the fold change followed by the p-value of the significance of the change for each pair of different time points (columns *week_1_week_10_change*, *week_1_week_10_pvalue* to *week_19_week_25_change*, *week_19_week_25_pvalue*).
(TSV)

**S7 Table. Changes in transcription factor binding to salinity and photoperiod.** Transcription factors (TF) with significant changes in global binding of transcription factor binding sites (TFBS) between time points representing a concerted change due to photoperiod conditions; week 1 (light) vs week 10 (dark), and week 10 (dark) vs 19 (light), or to salinity conditions; week 1 (fresh) vs week 25 (sea), and week 19 (fresh) vs week 25 (sea). Columns provide: TF name (*name*), time point comparison (*comparison*), p-value of the significance of the change between time points (*pvalue*), the fold change in different of TF binding scores (*change*), the conditions compared; photoperiod or salinity (*category*), the time point where positive change means more binding (*week_A*), the time point where negative change means more binding (*week_B*), and the conditions where there is significantly more binding of the TF (*sig*).
(TSV)

**S8 Table. Enrichment in binding patterns of transcription factor binding sites of genes in expression clusters.** Results of Fisher's exact tests for the enrichment in different binding patterns of transcription factor binding sites (TFBS) of genes in different expression clusters. Tested for each transcription factor (TF). Columns provide: the gene expression cluster (*deg_cluster*), the binding pattern tested made up of 4 digits representing the 4 time points in chronological order with *0* equating to the TFBS not bound while *1* is bound (*binding*), the total number of TFBS with the binding pattern associated by nearest proximity to genes within the expression cluster (*count*), p-value of Fisher's exact test (*pval*), odds ratio of test (*OR*), name of the TF motif for the TFBS (*TFBS_name*).
(TSV)

**S9 Table. Differentially methylated CpGs.** CpG sites significantly differentially methylated between time points of the experiment (FDR <0.05). Columns provide: Percentage score of the number of reads methylated at the CpG site for each time point (columns *week_1_score* to *week_25_score*), unique position of site in Atlantic salmon genome (*uniq_pos*), chromosome of site (*chr*), base position on chromosome (*locus*), $\log_2$ fold change in methylated read count across time points (columns *week_10_week_1_logFC* to *week_25_week_19_logFC*), average count of methylated reads across samples in $\log_2$ counts per million (log*CPM*), p-value for significance in change between any time points (*PValue*), false discovery rate adjusted p-value (*FDR*), associated gene's NCBI ID (*gene_id*), position of gene transcription start site (TSS) (*tss*), gene strand position (*strand*), distance of gene TSS to CpG site (*distance*), start and end positions of gene (*gene_start*, *gene_end*), gene width (*gene_width*), and type of genomic feature the CpG site is located in (*genomic_feature*).
(TSV)

**S1 File.**
(ZIP)

# Acknowledgments

Trond M. Kortner, from the Faculty of Veterinary Medicine, Norwegian University of Life Sciences, provided input during manuscript preparation and interpretation of results.

## Author Contributions

**Conceptualization:** Jon Olav Vik, Simen R. Sandve.

**Data curation:** Gareth B. Gillard, Fabian Grammes, Øystein Monsen.

**Funding acquisition:** Jon Olav Vik, Simen R. Sandve.

**Investigation:** Thomas N. Harvey, Line L. Røsæg.

**Methodology:** Thomas N. Harvey, Gareth B. Gillard, Line L. Røsæg, Fabian Grammes, Øystein Monsen, Torgeir R. Hvidsten.

**Supervision:** Simen R. Sandve.

**Visualization:** Thomas N. Harvey, Gareth B. Gillard, Line L. Røsæg.

**Writing – original draft:** Thomas N. Harvey, Gareth B. Gillard, Line L. Røsæg.

**Writing – review & editing:** Torgeir R. Hvidsten, Simen R. Sandve.

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
