## [Decision Letter · Decision Letter 0]

28 Nov 2023

PONE-D-23-31449The genome regulatory landscape of Atlantic salmon liver through smoltificationPLOS ONE

Dear Dr. Harvey,

Thank you for submitting your manuscript to PLOS ONE. After careful consideration, we feel that it has merit but does not fully meet PLOS ONE’s publication criteria as it currently stands. Therefore, we invite you to submit a revised version of the manuscript that addresses the points raised during the review process.

Your manuscript "The genome regulatory landscape of Atlantic salmon liver through smoltification" has been reviewed by to biologists that were in general favourable.

Main concerns we have relate more to representation and writing, not experiments, analyses or conclusions. Though some pointers may potentially lead to adjusted analyses and toned down conclusion(s).

The things we would like you to address are.

The summary of fold change (up vs down), patterns of expression etc (see rev 1).Disucss the limits of the study, only one tissue - liver- why important, 2 replicates only for ATAC, quality of gene annotations, other issues?Rev 1 raised concerns about the transition from global analyses to Lipid genes summary. Perhaps better highlight the functional diversity in this section, and better justify why focusing on lipids?Rev 1 calls for “Better explanation of methods, namely integrative analysis motif discovery and RNA-seq”, this can be accommodated in part with supplemental texts.Is there a technical reason for why timepoints 1 n 19 split from 10 n 25 on PC for the ATAC? Where all samples sequenced on same chip, or otherwise processed in parallel?Recommend adding a designated conclusion section. That summarizes better the molecular details (insights) in the context of the physiology of smoltification and the photoperiod question. The abstract could also gain from such editing, because in current state it ends with deep molecular focus, without referencing back to the main questions posed. ==============================

We look forward to receiving your revised manuscript.

Kind regards,

Arnar Palsson, Ph.D.

Academic Editor

PLOS ONE

Journal Requirements:

3. We are unable to open your Supporting Information file [Fig S1-3 and Table S1-9]. Please kindly revise as necessary and re-upload.

Additional Editor Comments:

Your manuscript "The genome regulatory landscape of Atlantic salmon liver through smoltification" has been reviewed by to biologists that were in general favourable.

Main concerns we have relate more to representation and writing, not experiments, analyses or conclusions. Though some pointers may potentially lead to adjusted analyses and toned down conclusion(s).

The things we would like you to address are.

1. The summary of fold change (up vs down), patterns of expression etc (see rev 1).

2. Disucss the limits of the study, only one tissue - liver- why important, 2 replicates only for ATAC, quality of gene annotations, other issues?

3. Rev 1 raised concerns about the transition from global analyses to Lipid genes summary. Perhaps better highlight the functional diversity in this section, and better justify why focusing on lipids?

4. Rev 1 calls for “Better explanation of methods, namely integrative analysis motif discovery and RNA-seq”, this can be accommodated in part with supplemental texts.

5. Is there a technical reason for why timepoints 1 n 19 split from 10 n 25 on PC for the ATAC? Where all samples sequenced on same chip, or otherwise processed in parallel?

6. Recommend adding a designated conclusion section. That summarizes better the molecular details (insights) in the context of the physiology of smoltification and the photoperiod question. The abstract could also gain from such editing, because in current state it ends with deep molecular focus, without referencing back to the main questions posed.

Reviewers' comments:

Reviewer's Responses to Questions

**Comments to the Author**

1. Is the manuscript technically sound, and do the data support the conclusions?

Reviewer #1: Yes

Reviewer #2: Yes

2. Has the statistical analysis been performed appropriately and rigorously? 

Reviewer #1: Yes

Reviewer #2: Yes

3. Have the authors made all data underlying the findings in their manuscript fully available?

Reviewer #1: No

Reviewer #2: Yes

4. Is the manuscript presented in an intelligible fashion and written in standard English?

Reviewer #1: Yes

Reviewer #2: Yes

5. Review Comments to the Author

Reviewer #1: I believe the datasets are relevant and quite unique for in salmon species. However, I would recommend a major revision to :

a) be more accurate when they mention up-down regulation (add log2FC) to be more specific

b) Perform a bit of extra-analysis in ATAC-seq before passing to motif discovery to have a global view of open-chromatin changes

c) Revise to provide a major angle (if there is margin to do so ) rather than focusing on lipid metabolism

d) improve the readership by better ligating the different sections of the manuscript. Also will help for non-salmon (smolt) experts

e) make sure data is fully available for the community (ATAC-seq)

f) Better explanation of methods, namely integrative analysis motif discovery and RNA-seq

Best Regards,

Marina

Reviewer #2: I would accept the article with minor revisions as outlined below:

The study presents the results of primary scientific research.

This is a very good primary scientific research article. Not only does it present RNA-seq, but also ATAC-seq and bisulfide sequencing results, which are very novel not just in Atlantic salmon but in most non-model species. The code has all been published, making this paper a valuable contribution to the field

Results reported have not been published elsewhere.

I could not find any of these results published anywhere, aside from its corresponding pre-print

Experiments, statistics, and other analyses are performed to a high technical standard and are described in sufficient detail.

The methods are clear; both the laboratory and bioinformatic pipelines are well-defined. The issue is that all the methodologies are used very independently. It's a shame that more attempts were not made to integrate the datasets; however, I would not consider this reason enough to not publish this work

Conclusions are presented in an appropriate fashion and are supported by the data.

The conclusion is that photoperiod does not seem to affect liver metabolism. This is presented appropriately, and all three different datasets seem well-described and analyzed. Although this is a negative result, it appears to be a good resource and an effective use of state-of-the-art omics. This work can be followed up on in the future

The article is presented in an intelligible fashion and is written in standard English.

Very well written, but could do with some proofreading, especially in the introduction.

The research meets all applicable standards for the ethics of experimentation and research integrity.

It does

The article adheres to appropriate reporting guidelines and community standards for data availability.

All data is publically available with accession numbers.

Comments

74-76 Although gill physiology has received most attention in the smoltification literature, other organs such as the liver also undergo large changes in function upon smoltification and seawater migration, with large implications for key metabolic traits. Please justify why the gill has received more research attention. This conclusion should be supported by references or data. Would be best to refer to it as being more studied that received more attention.

77 reared on different diets converges after smoltification [12, 13].

Not very clear what you mean by converges, can you explain it better?

81 Confounded does not make a lot of sense, maybe use another word?

94 consider changing a to the.

103 of the trial, but some mortality (8x fish) in one tank due to improper oxygenation after seawater transfer.

Would hypoxia not affect the results? Did you use this fish?

212-213 For each time point across the 25 week experiment group we generated two replicate samples of ATAC-seq data from the same livers sampled for RNA214 seq, at a depth of 55-72M reads.

Using only two samples is not statistically robust. Ideally, at least three replicates are recommended for ATAC-seq. Given the complexity of setting up this experiment, having more replicates may not be possible. However, having fewer replicates makes it difficult to infer statistically meaningful results with ATAC-seq across different time points. Integration of methods could have helped a bit with this.

Figures 2, 6, and 7 have low resolution and need improvement to make them clearer."

6. PLOS authors have the option to publish the peer review history of their article (what does this mean?). If published, this will include your full peer review and any attached files.

Reviewer #1: **Yes: **Marina Naval Sanchdz

Reviewer #2: No

---

## [Author Response · Author response to Decision Letter 0]

5 Feb 2024

Response to reviewers:

We thank the reviewers for their valuable feedback. Please find our point-by-point response to the reviewers’ comments below (response in blue). 

Editor comments

1. The summary of fold change (up vs down), patterns of expression etc (see rev 1).

We added a table of expression fold change values for lipid genes specifically (Table 1) to add more details, along with specific fold change value when we mention other gene expression change.

2. Disucss the limits of the study, only one tissue - liver- why important, 2 replicates only for ATAC, quality of gene annotations, other issues?

We have added some additional justification to the use of liver and the focus on lipid metabolism to the introduction (lines 76-81). Regarding using only two ATAC-seq replicates, we do not believe that this has impacted our analysis due to the reproducibility of the ATAC data in the PCA (Fig 5A). Also, using two replicates for ATAC-seq is common practice and has been established as a standard by the ENCODE consortium (https://www.encodeproject.org/data-standards/atac-seq/atac-encode4/). For this reason, we do not view having 2 replicates as a limit of the study and do not think it is necessary to add a discussion on this.

3. Rev 1 raised concerns about the transition from global analyses to Lipid genes summary. Perhaps better highlight the functional diversity in this section, and better justify why focusing on lipids?

We have added some justification for the focus on lipid metabolism to the introduction (lines 76 to 81). We also identified and corrected an error when describing the enrichment trends in cluster 4 and 5 in figure 2 (lines 135-138). Additionally, we have modified the discussion on metabolic remodeling to include amino acid and carbohydrate pathways since we also observe gene expression in these pathways decrease in freshwater smolts before seawater transition (lines 372 to 374).

4. Rev 1 calls for “Better explanation of methods, namely integrative analysis motif discovery and RNA-seq”, this can be accommodated in part with supplemental texts.

We reworded the explanation of this analysis in the relevant method section (line 582 to 589).

5. Is there a technical reason for why timepoints 1 n 19 split from 10 n 25 on PC for the ATAC? Where all samples sequenced on same chip, or otherwise processed in parallel?

All ATAC-seq samples were sequenced together and processed in parallel. We do not anticipate a technical reason for this. 

6. Recommend adding a designated conclusion section. That summarizes better the molecular details (insights) in the context of the physiology of smoltification and the photoperiod question. The abstract could also gain from such editing, because in current state it ends with deep molecular focus, without referencing back to the main questions posed.

We have added a dedicated conclusions section and modified the end of the abstract to tie the work together.

Reviewer 1

Reviewer #1: I believe the datasets are relevant and quite unique for in salmon species. However, I would recommend a major revision to :

a) be more accurate when they mention up-down regulation (add log2FC) to be more specific

We have added a table (table 1) which shows the log fold change of all the lipid metabolism genes mentioned in the results. Log2FC values for other genes mentioned in the results has been added to the text.

b) Perform a bit of extra-analysis in ATAC-seq before passing to motif discovery to have a global view of open-chromatin changes

We thank the reviewer for this suggestion as we believe it has greatly improved our analysis. To better link changes in chromatin accessibility to gene expression we have added three panels to figure 5 to highlight changes in chromatin accessibility during winter (weeks 1 to 10), during smoltification (weeks 10 to 19), and during seawater transfer (weeks 19 to 25) (Fig 5F). We then identified the nearest genes to these differentially accessible peaks and find that the direction of differential accessibility largely matches gene expression trends (Fig 5G and H). For example, peaks decreasing in accessibility during smoltification (week 10 to 19) were linked to genes belonging to cluster 7 which decreased in expression throughout the experiment and were enriched in lipid metabolism genes. We have added this information to the results section (lines 238 to 254) and discussion (lines 421 to 425)

c) Revise to provide a major angle (if there is margin to do so ) rather than focusing on lipid metabolism

Since we see smoltification associated downregulation of genes in amino acid and carbohydrate metabolism in addition to lipid metabolism, we have extended our interpretation to include these in preparatory metabolic remodeling (lines 372 to 374). However, we still emphasize lipid metabolism due to its relative importance in salmon farming. To support this we have added some additional justification for the focus on lipid metabolism to the introduction (lines 76-81)

d) improve the readership by better ligating the different sections of the manuscript. Also will help for non-salmon (smolt) experts

We have changed the subtitle of the second results section to better match the style of the other section subtitles.

e) make sure data is fully available for the community (ATAC-seq)

We have uploaded the ATAC-seq data to the European Nucleotide Archive under accession number PRJEB72206. Additionally these are now available through array express under accession numbers E-MTAB-11746 (RNA-seq) and E-MTAB-13743 (ATAC-seq)

f) Better explanation of methods, namely integrative analysis motif discovery and RNA-seq

We have modified the methods section to better explain the analysis (lines 582 to 589).

Reviewer 2

The study presents the results of primary scientific research.

This is a very good primary scientific research article. Not only does it present RNA-seq, but also ATAC-seq and bisulfide sequencing results, which are very novel not just in Atlantic salmon but in most non-model species. The code has all been published, making this paper a valuable contribution to the field

Results reported have not been published elsewhere.

I could not find any of these results published anywhere, aside from its corresponding pre-print

Experiments, statistics, and other analyses are performed to a high technical standard and are described in sufficient detail.

The methods are clear; both the laboratory and bioinformatic pipelines are well-defined. The issue is that all the methodologies are used very independently. It's a shame that more attempts were not made to integrate the datasets; however, I would not consider this reason enough to not publish this work

We have better integrated the data by adding three panels to figure 5. These show differentially accessible ATAC peaks and genes associated with these. We find that chromatin status is linked to changes in gene expression.

Conclusions are presented in an appropriate fashion and are supported by the data.

The conclusion is that photoperiod does not seem to affect liver metabolism. This is presented appropriately, and all three different datasets seem well-described and analyzed. Although this is a negative result, it appears to be a good resource and an effective use of state-of-the-art omics. This work can be followed up on in the future

The article is presented in an intelligible fashion and is written in standard English.

Very well written, but could do with some proofreading, especially in the introduction.

We have made several small grammatical corrections in the introduction.

The research meets all applicable standards for the ethics of experimentation and research integrity.

It does

The article adheres to appropriate reporting guidelines and community standards for data availability.

All data is publically available with accession numbers.

Comments

74-76 Although gill physiology has received most attention in the smoltification literature, other organs such as the liver also undergo large changes in function upon smoltification and seawater migration, with large implications for key metabolic traits. Please justify why the gill has received more research attention. This conclusion should be supported by references or data. Would be best to refer to it as being more studied that received more attention.

We have changed “received most attention” to “been most studied”. We also added “due to its role in osmoregulation” and reference to Nisembaum et al. 2021 to clarify why gill is more studied in relation to smoltification in salmon.

77 reared on different diets converges after smoltification [12, 13].

Not very clear what you mean by converges, can you explain it better?

We have edited this section to make this point more clear. It now reads as follows: “…reared on diets containing high or low long chain polyunstaturated fatty acids (LC-PUFA) have distinct lipid profiles during the parr stage, then during smoltification the lipid profiles converge”

81 Confounded does not make a lot of sense, maybe use another word?

To make this more clear and direct we removed “smolfication and seawater transfer were confounded (i.e…) ” and rather directly state that smolts in freshwater were not sampled.

94 consider changing a to the.

We have made this change.

103 of the trial, but some mortality (8x fish) in one tank due to improper oxygenation after seawater transfer.

Would hypoxia not affect the results? Did you use this fish?

We sampled 6 weeks after the hypoxia incident and assume there is no effect from this. Additionally, we have multiple tanks to account for such tank effects. We changed this to “immediately after seawater transfer” to emphasize that the mortality took place just after SW transfer and not closer to the time of sampling.

212-213 For each time point across the 25 week experiment group we generated two replicate samples of ATAC-seq data from the same livers sampled for RNA214 seq, at a depth of 55-72M reads.

Using only two samples is not statistically robust. Ideally, at least three replicates are recommended for ATAC-seq. Given the complexity of setting up this experiment, having more replicates may not be possible. However, having fewer replicates makes it difficult to infer statistically meaningful results with ATAC-seq across different time points. Integration of methods could have helped a bit with this.

While we agree that three ATAC-seq replicates may be more statistically robust, we do not believe that having two has impacted our results given the reproducibility of the ATAC data in the PCA plot. Additionally, at least two replicates has been established as the standard by the ENCODE consortium (https://www.encodeproject.org/data-standards/atac-seq/atac-encode4/). Since we are comparing ATAC and RNA signals, it is more important to have ATAC-seq and RNA-seq data coming from the same biological samples, which we have carried out for this analysis.

Figures 2, 6, and 7 have low resolution and need improvement to make them clearer.

We are unsure why the figures have low resolution. We have re-uploaded the figures as EPS files to attempt to resolve this issue.

Journal Requirements:

We have modified the title page to match the formatting PLOS one formatting style.

All data has been uploaded to ENA and Array express. Accession numbers have all been added to the data availability statement in the manuscript.

3. We are unable to open your Supporting Information file [Fig S1-3 and Table S1-9]. Please kindly revise as necessary and re-upload.

We are unsure why the supporting information could not be opened. We were able to open all supporting files from the PLOS one submission PDF file. To attempt to address this we have re-uploaded all supporting files in the revised submission and changed supplementary figure files to EPS format.

---

## [Decision Letter · Decision Letter 1]

11 Mar 2024

PONE-D-23-31449R1The genome regulatory landscape of Atlantic salmon liver through smoltificationPLOS ONE

Dear Dr. Harvey,

Thank you for submitting your manuscript to PLOS ONE. After careful consideration, we feel that it has merit but does not fully meet PLOS ONE’s publication criteria as it currently stands. Therefore, we invite you to submit a revised version of the manuscript that addresses the points raised during the review process.

This manuscript is in very good shape and represents a very impressive sets of data and results that are insightful and exciting. Unfortunately the critical reviewer was not available to comment on the MS again, so the editor opted for filling that role. There are a few general pointers, and numerous small editing suggestions, see below.

The larger points are:

Aspects of the experimental design are hard to glean from first pass reading. This includes both the text and layout of Figure 1 (see below) and analyses for figure 2. For the latter forn instance, is this just the comparison over the 4 timepoints in the “natural smoltification“ treatment (that is excluding the experimental treatments)?The results switch several times from past to present tense. Past tense is a more common style, but both can work. Do read over and check.The language in discussion does on occasion turn very assertive, but the fact is that the data are all correlative. I suggest adding a genera caveat on that, the data are correlative and while coherent do not represent formal experimental proof of said influence. Perhaps either in beginning or end of discussion.The wording in the methods, (particularly the ATAC) section, is quite challenging. I recommend careful proofreading for both content and style. Also related, was the ATAC data from a previously conducted study? (see wording in Line 581)

We look forward to receiving your revised manuscript.

Kind regards,

Arnar Palsson, Ph.D.

Academic Editor

PLOS ONE

Journal Requirements:

Additional Editor Comments:

This manuscript is in very good shape and represents a very impressive sets of data and results that are insightful and exciting. Unfortunately the critical reviewer was not available to comment on the MS again, so the editor opted for filling that role. There are a few general pointers, and numerous small editing suggestions, see below.

The larger points are:

1. Aspects of the experimental design are hard to glean from first pass reading. This includes both the text and layout of Figure 1 (see below) and analyses for figure 2. For the latter forn instance, is this just the comparison over the 4 timepoints in the “natural smoltification“ treatment (that is excluding the experimental treatments)?

2. The results switch several times from past to present tense. Past tense is a more common style, but both can work. Do read over and check.

3. The language in discussion does on occasion turn very assertive, but the fact is that the data are all correlative. I suggest adding a genera caveat on that, the data are correlative and while coherent do not represent formal experimental proof of said influence. Perhaps either in beginning or end of discussion.

4. The wording in the methods, (particularly the ATAC) section, is quite challenging. I recommend careful proofreading for both content and style. Also related, was the ATAC data from a previously conducted study? (see wording in Line 581)

Minor points and edits.

Line 96.

Reword. “test if the photoperiodic history is a major factor impacts” to “test if the photoperiodic history influences…” to

Line 110 and Figure 1. Legend.

Add some details. How many fish where reared in each tank, treatment combniation? How many timepoint/treatment combinations were sampled? How many where sampled for each method (Same fish sampled for both RNAseq and ATAC seq?) Clarify that the fishes reprsented are a random sample of the fish in tank/treatment. Add statistics on the size data (either the represented fish or the tank/treatment as a whole).

Line 132.

Maybe specify that edgeR is used “using an ANOVA-like test.”,. comparing the 4 or 7? experimental groups?

Line 134

Can drop “major”.

Line 139.

Can you invert the order “To associate well defined metabolic or signaling processes to the different gene expression trends,”

“To study the function of differentially expressed clusters we…”

Line 141.

Why not start with cluster 1? Or rename them so they appear in order in text. Cluster 6 is not talked about.

Figure 2. Legend.

“Global gene expression changes across life-stage.” Title can be more descriptive. Can apply to any organisms really.

Legend indicates 6 clusters, figure has 7.

Line 164.

Did all the genes in this lipid metabolism pathway have significantly altered expression? Were some not DE?

Line 200

Italic “de novo”

Line 210

Consider placing the fish earlier in the sentence, it is the organism in focus “expression suggests a reduced capacity to synthesize fatty acids in liver of fish after transition to sea,”

Figure 3 legend.

Does expression of all these genes differ signifcantly with time?

Line 234.

Check wording “were concentrated were called to represent regions” – maybe “defined as”?

Line 254

Check wording “with the nearest gene and counted the distribution of previously identified”

Line 249

Impressive improvement of the representation and cool results. P values from statistical tests are however missing for many of those. Can be incorporated into text or figure 5 legend.

Line 279

Drop “therefore”

Figure 6 legend.

The “regulation” variable in the Figure is poorly explained. Add details to legend. “Green: … ColourX:… “

Line 287

Add the caveat that you are inferring TF function based on data from other Eukaryotes / metazoa.

Line 309

Rephrase “Next, we wanted to link TF binding patterns to the specific gene” perhaps.

“Next we tested for a link between TF binding patterns and gene…”

Line 320.

Check wording “had nearby TFBS with NFIX binding” maybe “had TFBS most likely bound by NFIX”??

Line 336.

“Cluster 6 was not enriched”

Line 355

“To investigate the role DNA methylation had on the gene expression changes” rephrase, ask first IF it has a role.

Line 356.

What is the average coverage in the genome?

Line 355.

Several sentences start with numbers in this paragraph. Rephrase.

Line 378

Rephrase “An important feature of smoltification is how the process prepares” maybe “An important question about S is how…”

Line 381

What kind of substance is “NKA a1b”? Does this show up in the liver?

Line 383.

What happens to lipid metabolism. Goes up or down?

Lin 415

Check statement of First “Yet, no genome wide studies of DNA…” can we reworded. “to our knowledge published”??

Line 436

Check wording, something off “with a decrease in peak accessibility of regions near many of these genes”

Line 474

Add liver in the sentence somewhere.

Line 481

Rephrase “history likely does not play a major role” maybe “history is unlikely to play a major role”

Line 482

Reword “contrast from studies in gill.”

Line 503.

Reword. “Sampling began 21 weeks after first 503 feeding, here called week 1”

Line 523

“of at least eight was used to make RNA-seq libraries” How many samples failed this test?

Line 530

“read alignments “ to “alignment of reads”

Line 533.

Provide hyperlink nf-co.re/rnaseq.

Line 539

Strange wording, fix “Differentially expressed genes (DEGs) were tested for first differences across all experimental groupS?”

Line 542

Strange wording, fix “DEGs were chosen from the results using an FDR cutoff of”

Line 542

“Euclidean distances” calculated with what?

Line 546

Check wording ”DEGs were also tested between experimental and control groups”

Line 553

Was this done directly when sample taken, or retrieved from RNA later? “Two replicate liver tissue samples were used …”

Line 567

Reference for “nf-core atacseq pipeline (v1.2.1),”?

Line 571

Wording “computed through the pipeline” is a bit peculiar. Can you rephrase. Phrase used elsewhere also.

Line 565

Check wording. “combined with the consensus annotated peak file from the pipeline, annotating peaks to genes by the shortest distance

to gene TSS.” In general, check wording of all newly added segments. They tend to be the most troublesome. Like the following sentence “We further associated with the differential accessible peaks the set of DEGs determined”

Line 581

Reword “TF footing in the unified ATAC peak set previously generated was done using”

Line 584

During the “protocol”?? “locally blocking transposase activity during the ATAC protocol.”

Line 586

Reword “o associate these footprints with specific TFs.”

Line 588

Perfect for supplemental data that other researchers could use. “(generated in-house).”

Line 596.

How was the overlap tested “We tested…”? Which statistical environment, how automated over the whole dataset?

Line 608

“Livers from four fish per time point (three fish for week 25) were” Are these samples from the same fish as the RNA and ATAC seq where done on? Indicate in results and in beginning of these sections if these are the same or not.

Reviewers' comments:

Reviewer's Responses to Questions

**Comments to the Author**

1. If the authors have adequately addressed your comments raised in a previous round of review and you feel that this manuscript is now acceptable for publication, you may indicate that here to bypass the “Comments to the Author” section, enter your conflict of interest statement in the “Confidential to Editor” section, and submit your "Accept" recommendation.

Reviewer #2: All comments have been addressed

2. Is the manuscript technically sound, and do the data support the conclusions?

Reviewer #2: Yes

3. Has the statistical analysis been performed appropriately and rigorously? 

Reviewer #2: Yes

4. Have the authors made all data underlying the findings in their manuscript fully available?

Reviewer #2: Yes

5. Is the manuscript presented in an intelligible fashion and written in standard English?

Reviewer #2: Yes

6. Review Comments to the Author

Reviewer #2: (No Response)

7. PLOS authors have the option to publish the peer review history of their article (what does this mean?). If published, this will include your full peer review and any attached files.

Reviewer #2: No

---

## [Author Response · Author response to Decision Letter 1]

22 Mar 2024

Response to reviewers:

We thank the editor for their valuable feedback. Please find our point-by-point response to the comments below (response in blue). 

Editor comments

1. Aspects of the experimental design are hard to glean from first pass reading. This includes both the text and layout of Figure 1 (see below) and analyses for figure 2. For the latter forn instance, is this just the comparison over the 4 timepoints in the “natural smoltification“ treatment (that is excluding the experimental treatments)?

Figure 1 has been changed with a diagram to better describe the experimental timeline. We removed the use of experimental and control group terminology throughout and replaced with more explicit descriptions of the sample groups (see comment 4).

2. The results switch several times from past to present tense. Past tense is a more common style, but both can work. Do read over and check.

We have read through the results section and changed present to past tense where appropriate.

3. The language in discussion does on occasion turn very assertive, but the fact is that the data are all correlative. I suggest adding a genera caveat on that, the data are correlative and while coherent do not represent formal experimental proof of said influence. Perhaps either in beginning or end of discussion.

We have added a caveat at the end of the discussion extending to future work.

4. The wording in the methods, (particularly the ATAC) section, is quite challenging. I recommend careful proofreading for both content and style. Also related, was the ATAC data from a previously conducted study? (see wording in Line 581)

We have read through the entire methods section and clarified wording where appropriate. More specifically, wording around the different groups in the feeding trial (previously experimental and control groups) was changed to be more explicit. We now refer to the groups with different photoperiods as short and long photoperiod groups and the fish that stayed in freshwater as the freshwater control group. This change has been implemented throughout the paper.

The data referred to on line 581 was data generated in the earlier section, not from another study. We have removed “previously generated” to avoid this confusion.

Minor points and edits.

Line 96.

Reword. “test if the photoperiodic history is a major factor impacts” to “test if the photoperiodic history influences…” to

We have made this change.

Line 110 and Figure 1. Legend.

Add some details. How many fish where reared in each tank, treatment combniation? How many timepoint/treatment combinations were sampled? How many where sampled for each method (Same fish sampled for both RNAseq and ATAC seq?) Clarify that the fishes reprsented are a random sample of the fish in tank/treatment. Add statistics on the size data (either the represented fish or the tank/treatment as a whole).

We have added details to the text and to the figure caption. We have also reorganized the figure to provide more information.

Line 132.

Maybe specify that edgeR is used “using an ANOVA-like test.”,. comparing the 4 or 7? experimental groups?

We have reworded this line to make it clear that the ANOVA-like test is being performed on the fish experiencing short photoperiod and seawater transfer.

Line 134

Can drop “major”.

We have dropped the word “major”.

Line 139.

Can you invert the order “To associate well defined metabolic or signaling processes to the different gene expression trends,”

“To study the function of differentially expressed clusters we…”

We have implemented this suggestion.

Line 141.

Why not start with cluster 1? Or rename them so they appear in order in text. Cluster 6 is not talked about.

We have moved the sentence about cluster 1 to the beginning and added a sentence on cluster 6 stating that no pathways were significantly enriched.

Figure 2. Legend.

“Global gene expression changes across life-stage.” Title can be more descriptive. Can apply to any organisms really.

Legend indicates 6 clusters, figure has 7.

We has added salmon to the figure legend title and fixed the number of clusters in the text.

Line 164.

Did all the genes in this lipid metabolism pathway have significantly altered expression? Were some not DE?

We made additions to Table 1 and Figure 3 to indicate the lipid genes that were also tested to be differentially expressed across in the timepoints (Figure 2 results).

Line 200

Italic “de novo”

We have fixed this.

Line 210

Consider placing the fish earlier in the sentence, it is the organism in focus “expression suggests a reduced capacity to synthesize fatty acids in liver of fish after transition to sea,”

We have changed fish to salmon to be more specific and moved it earlier in the sentence.

Figure 3 legend.

Does expression of all these genes differ signifcantly with time?

We added annotation of which genes are significantly different across timepoints, as mentioned above.

Line 234.

Check wording “were concentrated were called to represent regions” – maybe “defined as”?

We have changed “called to represent” to “defined as”.

Line 254

Check wording “with the nearest gene and counted the distribution of previously identified”

We have reworded this slightly to make it clearer.

Line 249

Impressive improvement of the representation and cool results. P values from statistical tests are however missing for many of those. Can be incorporated into text or figure 5 legend.

P-value cutoff for differentially accessible peaks (Fig 5F) is mentioned in the figure legend (FDR < 0.05). Marked proportions in Fig 5G that are significantly greater by Fisher’s exact test p-value <0.05 by asterisks, with note in legend.

Line 279

Drop “therefore”

We made this deletion.

Figure 6 legend.

The “regulation” variable in the Figure is poorly explained. Add details to legend. “Green: … ColourX:… “

We have added text to the figure legend to explicitly state which colors correspond to which environmental variables.

Line 287

Add the caveat that you are inferring TF function based on data from other Eukaryotes / metazoa.

We have added “in mammals” to this sentence because the references we use are from studies in mammals.

Line 309

Rephrase “Next, we wanted to link TF binding patterns to the specific gene” perhaps.

“Next we tested for a link between TF binding patterns and gene…”

We have made this change.

Line 320.

Check wording “had nearby TFBS with NFIX binding” maybe “had TFBS most likely bound by NFIX”??

We worded to be explicit it is NFIX motifs that are measured to be differentially bound.

Line 336.

“Cluster 6 was not enriched”

We have made this change.

Line 355

“To investigate the role DNA methylation had on the gene expression changes” rephrase, ask first IF it has a role.

We have changed this to “To investigate if DNA methylation played a role in the gene expression changes”.

Line 356.

What is the average coverage in the genome?

We added details on the consensus set of CpGs which included minimal read coverage of 10x.

Line 355.

Several sentences start with numbers in this paragraph. Rephrase.

We have revised these sentences.

Line 378

Rephrase “An important feature of smoltification is how the process prepares” maybe “An important question about S is how…”

This first statement is describing the preparatory nature of smoltification, not asking the question of how it works. This has been revised to be phrased less as a question and more as a statement.

Line 381

What kind of substance is “NKA a1b”? Does this show up in the liver?

This is the Na+/K+ ATPase α1b isoform which pumps ions out of the gills. This does not play the same role in the liver.

Line 383.

What happens to lipid metabolism. Goes up or down?

We are unsure what this comment refers to. In the same sentence it is stated “lipid metabolism related gene expression is known to decrease in liver of seawater stage Atlantic salmon”.

Line 415

Check statement of First “Yet, no genome wide studies of DNA…” can we reworded. “to our knowledge published”??

We have changed this to “Yet, to our knowledge no genome wide studies of DNA methylation, chromatin accessibility, and TFs involved in driving these transcriptional and physiological changes have been published”.

Line 436

Check wording, something off “with a decrease in peak accessibility of regions near many of these genes”

Changed “was associated” to “coincided”.

Line 474

Add liver in the sentence somewhere.

Added “in liver of Atlantic salmon”.

Line 481

Rephrase “history likely does not play a major role” maybe “history is unlikely to play a major role”

We have made this change.

Line 482

Reword “contrast from studies in gill.”

Changed to “unlike”.

Line 503.

Reword. “Sampling began 21 weeks after first 503 feeding, here called week 1”

We have made this change.

Line 523

“of at least eight was used to make RNA-seq libraries” How many samples failed this test?

No samples had a RIN less than 8. This is the minimum quality of RNA used in this study.

Line 530

“read alignments “ to “alignment of reads”

We have made this change.

Line 533.

Provide hyperlink nf-co.re/rnaseq.

We have added a hyperlink.

Line 539

Strange wording, fix “Differentially expressed genes (DEGs) were tested for first differences across all experimental groups?”

Removed “first” and changed to “short photoperiod group time points”.

Line 542

Strange wording, fix “DEGs were chosen from the results using an FDR cutoff of”

Changed to “DEGs were found using an FDR cutoff of”.

Line 542

“Euclidean distances” calculated with what?

We have added the R function used.

Line 546

Check wording ”DEGs were also tested between experimental and control groups”

Change this to “Differential expression analysis was also performed between”.

Line 553

Was this done directly when sample taken, or retrieved from RNA later? “Two replicate liver tissue samples were used …”

The liver samples were immediately processed after sampled for each timepoint. The text has been revised to make this clear.

Line 567

Reference for “nf-core atacseq pipeline (v1.2.1),”?

Added the DOI to the version. Did the same for the RNAseq pipeline.

Line 571

Wording “computed through the pipeline” is a bit peculiar. Can you rephrase. Phrase used elsewhere also.

Changed “computed through” to “obtained from”.

Line 565

Check wording. “combined with the consensus annotated peak file from the pipeline, annotating peaks to genes by the shortest distance to gene TSS.” In general, check wording of all newly added segments. They tend to be the most troublesome. Like the following sentence “We further associated with the differential accessible peaks the set of DEGs determined”

This has been reworded to improve clarity.

Line 581

Reword “TF footing in the unified ATAC peak set previously generated was done using”

We have slightly reworded this sentence.

Line 584

During the “protocol”?? “locally blocking transposase activity during the ATAC protocol.”

This sentence has been restructured to be more clear.

Line 586

Reword “o associate these footprints with specific TFs.”

Changed to “to associate identify TFs associated with these footprints.”

Line 588

Perfect for supplemental data that other researchers could use. “(generated in-house).”

We used RepeatMasker to mask simple repetitive regions in the genome from the analysis. We have updated the methods section to reflect this.

Line 596.

How was the overlap tested “We tested…”? Which statistical environment, how automated over the whole dataset?

We used a Fisher’s exact test. This is already mentioned in the paragraph (line 608) so we have not modified the text.

Line 608

“Livers from four fish per time point (three fish for week 25) were” Are these samples from the same fish as the RNA and ATAC seq where done on? Indicate in results and in beginning of these sections if these are the same or not.

These were not the same fish, but they were sampled at the same time as the fish used for RNAseq and ATACseq. We have added this detail to the methods and results sections.

---

## [Editor Report · Decision Letter 2]

3 Apr 2024

The genome regulatory landscape of Atlantic salmon liver through smoltification

PONE-D-23-31449R2

Dear Dr. Harvey,

We’re pleased to inform you that your manuscript has been judged scientifically suitable for publication and will be formally accepted for publication once it meets all outstanding technical requirements.

Kind regards,

Arnar Palsson, Ph.D.

Academic Editor

PLOS ONE
---

## [Editor Report · Acceptance letter]

5 Apr 2024

PONE-D-23-31449R2 

PLOS ONE

Dear Dr. Harvey, 

I'm pleased to inform you that your manuscript has been deemed suitable for publication in PLOS ONE. Congratulations! Your manuscript is now being handed over to our production team.

Kind regards, 

on behalf of

Dr. Arnar Palsson 

Academic Editor

PLOS ONE